# Towards Better Understanding of Training Certifiably Robust Models against Adversarial Examples

**Sungyoon Lee**
Korea Institute for Advanced Study (KIAS)
sungyoonlee@kias.re.kr

**Woojin Lee**
Dongguk University-Seoul
wj926@dgu.ac.kr

**Jinseong Park**
Seoul National Univeristy
jinseong@snu.ac.kr

**Jaewook Lee**
Seoul National University
jaewook@snu.ac.kr

## Abstract

We study the problem of training certifiably robust models against adversarial examples. Certifiable training minimizes an upper bound on the worst-case loss over the allowed perturbation, and thus the tightness of the upper bound is an important factor in building certifiably robust models. However, many studies have shown that Interval Bound Propagation (IBP) training uses much looser bounds but outperforms other models that use tighter bounds. We identify another key factor that influences the performance of certifiable training: *smoothness of the loss landscape*. We find significant differences in the loss landscapes across many linear relaxation-based methods, and that the current state-of-the-arts method often has a landscape with favorable optimization properties. Moreover, to test the claim, we design a new certifiable training method with the desired properties. With the tightness and the smoothness, the proposed method achieves a decent performance under a wide range of perturbations, while others with only one of the two factors can perform well only for a specific range of perturbations. Our code is available at https://github.com/sungyoon-lee/LossLandscapeMatters.

## 1 Introduction

Despite the success of deep learning in many applications, the existence of adversarial example, an imperceptibly modified input that is designed to fool the neural network [34, 4], hinders the application of deep learning to safety-critical domains. There has been increasing interest in building a model that is robust to adversarial attacks [14, 26, 23, 47, 43, 16, 41]. However, most defense methods evaluate their robustness with adversarial accuracy against predefined attacks. Thus, these defenses can be broken by new attacks [1, 6, 37, 8, 9].

To this end, many training methods have been proposed to build a certifiably robust model that can be guaranteed to be robust to adversarial perturbations [17, 28, 39, 11, 24, 15, 46, 2, 19]. They develop an upper bound on the worst-case loss over valid adversarial perturbations and minimize it to train a certifiably robust model. These certifiable training methods can be mainly categorized into two types: linear relaxation-based methods and bound propagation methods. Linear relaxation-based methods use relatively tighter bounds, but are slow, hard to scale to large models, and memory-inefficient [39, 40, 11]. On the other hand, bound propagation methods, represented by Interval Bound Propagation (IBP), are fast and scalable due to the use of simple but much looser bounds [24, 15]. One would expect that training with tighter bounds would lead to better performance, but IBP outperforms linear relaxation-based methods, especially when the perturbation is large, despite using much looser bounds.

35th Conference on Neural Information Processing Systems (NeurIPS 2021).

These observations on the performance of certifiable training methods raise the following questions:

*Why does certifiable training with tighter bounds not result in a better performance?*
*What other factors may influence the performance of certifiable training?*

In this paper, we provide empirical and theoretical analysis to answer these questions. First, we demonstrate that IBP [15] has a more favorable (smooth) loss landscape than other linear relaxation-based methods, and thus it often leads to better performance even with much looser bounds. To account for this difference, we present a unified view of IBP and linear relaxation-based methods and find that the relaxed gradient approximation (which will be defined in Definition 1) of each method plays a crucial role in its optimization behavior. Based on the analysis of the loss landscape and the optimization behavior, we propose a new certifiable training method that has a favorable loss landscape with tighter bounds. As a result, the proposed method can achieve a decent performance under a wide range of perturbations. We summarize the contributions of this study as follows:

- We provide empirical and theoretical analysis of the loss landscape of certifiable training methods and find that smoothness of the loss landscape is important for building certifiably robust models, in addition to the tightness of the upper bound.
- We find that the *relaxed gradient approximation* of a certifiable training method plays a major role in shaping the loss landscape, determining its optimization behavior.
- To verify our claims, we propose a certifiable training method with tighter bounds and a favorable loss landscape. With the two key factors, the proposed method can achieve a decent performance under a wide range of perturbations, while others with only one of the two can perform well only for a specific range of the adversarial perturbations.

## 2 Related Work

Earlier studies on training certifiably robust models were limited to 2-layered networks [17, 27]. To scale to larger networks, a line of work has proposed the use of linear relaxation of nonlinear activation to formulate a robust optimization. Then, a dual problem is considered and a dual feasible solution is used to simplify the computation further. By doing so, Wong and Kolter [39] built a method that can scale to a 4-layered network, and later, Wong et al. [40] used Cauchy random projections to scale to much larger networks. However, they are still slow and memory-inefficient. Dvijotham et al. [11] proposed a method called predictor-verifier training (PVT), which uses a verifier network to optimize the dual solution. This is similar to our proposed method but we do not require any additional network. Similarly, a recent work called FROWN [22] used an optimizer to obtain tighter robustness certificates, but it focuses on efficient certification, not on certifiable training. Xiao et al. [42] proposed to add regularization technique with adversarial training for inducing ReLU stability, but it is less effective than other certified defenses.

Mirman et al. [24] proposed the propagation of a geometric bound (called domain) through the network to yield an outer approximation in logit space. This can be done with an efficient layerwise computation that exploits interval arithmetic. Over the outer domain, one can compute the worst-case loss to be minimized during training. Gowal et al. [15] used a special case of the domain propagation called Interval Bound Propagation (IBP) using the simplest domain, the interval domain (or interval bound). They also introduced a different objective function, heuristic scheduling on the hyperparameters, and elision of the last layer to stabilize the training and to improve the performance.

Both approaches, linear relaxation-based methods and bound propagation methods, use an upper bound on the worst-case loss. Bound propagation methods exploit much looser upper bounds, but they enjoy an unexpected benefit in many cases: better robustness than linear relaxation-based methods. Balunovic and Vechev [2] hypothesized that the complexity of the loss computation makes the optimization more difficult, which could be a reason why IBP outperforms linear relaxation-based methods. They proposed a new optimization procedure with the existing linear relaxation. In this paper, we further investigate the causes of the difficulties in the optimization. Recently, Zhang et al. [46] proposed CROWN-IBP which uses linear relaxation in a verification method called CROWN [45] in conjunction with IBP to train a certifiably robust model.

Although beyond our focus here, there are other lines of work on certification with convex relaxation [38, 31, 33, 32, 45, 5, 29, 22, 35, 10, 48] and probabilistic certification called randomized smoothing

[20, 18, 7, 44, 25]. They focus solely on the tightness of the certified bounds. However, our work studies certifiable training from an optimization perspective.

# 3 Background

First, we provide a brief overview of certifiable training methods. Then, we consider IBP [15] as a special case of linear relaxation-based methods. This unified view on certifiable training methods helps us to comprehensively analyze the differences between the two approaches: bound propagation and linear relaxation. We present the details of IBP in Appendix B.

## 3.1 Notations and Certifiable Training

We consider a $c$-class classification problem with a neural network $f(\boldsymbol{x}; \boldsymbol{\theta})$ with the layerwise operations $\boldsymbol{z}^{(k)} = h^{(k)}(\boldsymbol{z}^{(k-1)})$ $(k = 1, \cdots, K)$ and the input $\boldsymbol{z}^{(0)} = \boldsymbol{x} \in \mathcal{X}$. The corresponding probability function is denoted by $\boldsymbol{p}_f = \text{softmax} \circ f : \mathcal{X} \to [0,1]^c$ with subscript $f$. We denote a subnetwork with $k$ operations as $h^{[k]} = h^{(k)} \circ \cdots \circ h^{(1)}$. For a linear operation $h^{(k)}$, we use $\boldsymbol{W}^{(k)}$ and $\boldsymbol{b}^{(k)}$ to denote the weight and the bias for the layer. We consider the robustness of the classifier against the norm-bounded perturbation set $\mathbb{B}(\boldsymbol{x}, \epsilon) = \{\boldsymbol{x}' \in \mathcal{X} : \|\boldsymbol{x}' - \boldsymbol{x}\| \leq \epsilon\}$ with the perturbation level $\epsilon$. Here, we mainly focus on the $\ell_\infty$-norm bounded set. To compute the margin between the true class $y$ for the input $\boldsymbol{x}$ and the other classes, we define a $c \times c$ matrix $\boldsymbol{C}(y) = \boldsymbol{I} - \boldsymbol{1}e^{(y)^T}$ with $(\boldsymbol{C}(y)\boldsymbol{z}^{(K)})_m = \boldsymbol{z}_m^{(K)} - \boldsymbol{z}_y^{(K)}$ $(m = 0, \cdots, c-1)$. For the last linear layer, the weights $\boldsymbol{W}^{(K)}$ and the bias $\boldsymbol{b}^{(K)}$ are merged with $\boldsymbol{C}(y)$, that is, $\boldsymbol{W}^{(K)} \equiv \boldsymbol{C}(y)\boldsymbol{W}^{(K)}$ and $\boldsymbol{b}^{(K)} \equiv \boldsymbol{C}(y)\boldsymbol{b}^{(K)}$, yielding the margin score function $\boldsymbol{s}(\boldsymbol{x}, y; \boldsymbol{\theta}) = \boldsymbol{C}(y)f(\boldsymbol{x}; \boldsymbol{\theta}) = f(\boldsymbol{x}; \boldsymbol{\theta}) - f_y(\boldsymbol{x}; \boldsymbol{\theta})\boldsymbol{1}$ satisfying $\boldsymbol{p_s} = \boldsymbol{p}_f$. Then we can define the worst-case margin score $\boldsymbol{s}^*(\boldsymbol{x}, y, \epsilon; \boldsymbol{\theta}) = \max_{\boldsymbol{x}' \in \mathbb{B}(\boldsymbol{x}, \epsilon)} \boldsymbol{s}(\boldsymbol{x}', y; \boldsymbol{\theta})$ where $\max$ is element-wise maximization. With an upper bound $\overline{\boldsymbol{s}}$ on the worst-case margin score, $\overline{\boldsymbol{s}} \geq \boldsymbol{s}^*$, we can provide an upper bound on the worst-case loss over valid adversarial perturbations as follows:

$$\mathcal{L}(\overline{\boldsymbol{s}}(\boldsymbol{x}, y, \epsilon; \boldsymbol{\theta}), y) \geq \max_{\boldsymbol{x}' \in \mathbb{B}(\boldsymbol{x}, \epsilon)} \mathcal{L}(f(\boldsymbol{x}'; \boldsymbol{\theta}), y) \tag{1}$$

for cross-entropy loss $\mathcal{L}$ [39]. We denote the empirical loss $\mathbb{E}_{(\boldsymbol{x}, y)}[\mathcal{L}(\overline{\boldsymbol{s}}(\boldsymbol{x}, y, \epsilon; \boldsymbol{\theta}), y)]$ as $\mathcal{L}^\epsilon(\boldsymbol{\theta})$, Therefore, we can formulate certifiable training as a minimization of the upper bound, $\min_{\boldsymbol{\theta}} \mathcal{L}^\epsilon(\boldsymbol{\theta})$, instead of directly solving $\min_{\boldsymbol{\theta}} \mathbb{E}_{(\boldsymbol{x}, y)}[\max_{\boldsymbol{x}' \in \mathbb{B}(\boldsymbol{x}, \epsilon)} \mathcal{L}(f(\boldsymbol{x}'; \theta), y)]$ which is infeasible. Note that adversarial training [23] uses a strong iterative gradient-based attack (PGD) to provide a lower bound on the worst-case loss to be minimized, but it cannot provide a certifiably robust model. Whenever possible, we will simplify the notations by omitting variables such as $\boldsymbol{x}, y, \epsilon$, and $\boldsymbol{\theta}$.

## 3.2 Linear Relaxation-based Methods

For a subnetwork $h^{[k]}$, given with the pre-activation upper/lower bounds, $\boldsymbol{u}$ and $\boldsymbol{l}$, for each nonlinear activation function $h$ in $h^{[k]}$, linear relaxation-based methods [39, 40, 46] use a relaxation of the activation function by two elementwise linear function bounds, $\overline{h}$ and $\underline{h}$, that is, $\underline{h}(\boldsymbol{z}) \leq h(\boldsymbol{z}) \leq \overline{h}(\boldsymbol{z})$ for $\boldsymbol{l} \leq \boldsymbol{z} \leq \boldsymbol{u}$. We denote the function bounds as $\overline{h}(\boldsymbol{z}) = \overline{\boldsymbol{a}} \odot \boldsymbol{z} + \overline{\boldsymbol{b}}$ and $\underline{h}(\boldsymbol{z}) = \underline{\boldsymbol{a}} \odot \boldsymbol{z} + \underline{\boldsymbol{b}}$ for some $\overline{\boldsymbol{a}}, \overline{\boldsymbol{b}}, \underline{\boldsymbol{a}}$, and $\underline{\boldsymbol{b}}$, where $\odot$ denotes the elementwise (Hadamard) product. Using all the function bounds $\overline{h}$'s and $\underline{h}$'s for the nonlinear activations in conjunction with the linear operations in $h^{[k]}$, an $i$th (scalar) activation $h_i^{[k]}(\cdot) \in \mathbb{R}$ can be upper bounded by a linear function $\overline{h}_i^{[k]}(\cdot) = \boldsymbol{g}^T \cdot + b$ over $\mathbb{B}(\boldsymbol{x}, \epsilon)$ as in Zhang et al. [45]. This can be equivalently explained with the dual relaxation viewpoint in Wong and Kolter [39]. Further details are provided in Appendix C. Now we are ready to upper bound the activation $h_i^{[k]}$ over $\mathbb{B}(\boldsymbol{x}, \epsilon)$.

**Definition 1** (Linear Relaxation with Relaxed Gradient Approximation). *For each neuron activation $h_i^{[k]}$, a linear relaxation method computes an upper approximation of the activation over $\mathbb{B}(\boldsymbol{x}, \epsilon)$ by using $\boldsymbol{g} \in \mathbb{R}^d$ and $b \in \mathbb{R}$ as follows:*

$$\max_{\boldsymbol{x}' \in \mathbb{B}(\boldsymbol{x}, \epsilon)} h_i^{[k]}(\boldsymbol{x}') \leq \max_{\boldsymbol{x}' \in \mathbb{B}(\boldsymbol{x}, \epsilon)} \boldsymbol{g}^T \boldsymbol{x}' + b = \boldsymbol{g}^T \boldsymbol{x} + \epsilon \|\boldsymbol{g}\|_* + b. \tag{2}$$

*We call $\boldsymbol{g}$ the relaxed gradient approximation of $h_i^{[k]}$ over $\mathbb{B}(\boldsymbol{x}, \epsilon)$.*

Similarly, we can obtain the corresponding lower bound. Inductively using these upper/lower bounds on the output of the subnetwork, we can obtain the bounds for the next subnetwork $h^{[k+1]}$ and then for the whole network $s$. The final bound $\overline{s}$ on the whole network $s$ can then be used in the objective (1). The tightness of the bounds $\overline{s}$ and $\mathcal{L}(\overline{s}, y)$ highly depend on how the linear bounds $\overline{h}$ and $\underline{h}$ are chosen in each layer.

**Unified view of IBP and linear relaxation-based methods**  IBP can also be considered as a linear relaxation-based method using zero-slope ($\overline{a} = \underline{a} = 0$) linear bounds, $\overline{h}(z) = u^+$ and $\underline{h}(z) = l^+$, where $v^+ = \max(v, 0)$ and $v^- = \min(v, 0)$. Thus, the bounds of a nonlinear activation depend only on the pre-activation bounds $u$ and $l$ for the activation layer, substantially reducing the feed-forward/backpropagation computations. CROWN-IBP [46] applies different linear relaxation schemes to the subnetworks and the whole network. It uses the same linear bounds as IBP for the subnetworks $h^{[k]}$ for $k < K$ except for the network $s = h^{[K]}$ itself, and uses $\overline{h}(z) = \frac{u^+}{u^+ - l^=} \odot (z - l^-)$ and $\underline{h}(z) = \mathbf{1}[u^+ + l^- > 0] \odot z$ for the whole network $s$. Moreover, CROWN-IBP uses interpolations between two bounds with the mixing weight $\beta$, the IBP bound and the CROWN-IBP bound, with the following objective:

$$\mathcal{L}\left((1 - \beta)\overline{s}^{\text{IBP}}(x, y, \epsilon; \theta) + \beta\overline{s}^{\text{CROWN-IBP}}(x, y, \epsilon; \theta), y\right). \tag{3}$$

CAP [39, 40] uses the linear bounds $\overline{h}(z) = \frac{u^+}{u^+ - l^=} \odot (z - l^-)$ and $\underline{h}(z) = \frac{u^+}{u^+ - l^=} \odot z$ for all subnetworks $h^{[k]}$ and the entire network. As CAP utilizes the linear bounds for each neuron, it is slow and memory-inefficient. It can be shown that tighter relaxations on nonlinear activations yield a tighter bound on the worst-case margin score $s^*$. To specify the linear relaxation variable $\phi \equiv \{\overline{a}, \underline{a}, \overline{b}, \underline{b}\}$ used in relaxation, we use the notation $\overline{s}(x, y, \epsilon; \theta, \phi)$ with $\phi$. CROWN-IBP and CAP generally yield a much tighter bound than IBP. These relaxation schemes are illustrated in Appendix D.

## 4  What factors influence the performance of certifiable training?

One would expect that a tighter upper bound on the worst-case loss in (1) is beneficial in certifiable training. However, several previous works have shown that this is not the case: IBP performs better than linear relaxation-based methods in many cases while utilizing a much looser bound. We investigate the loss landscape and the optimization behavior of IBP and other linear relaxation-based methods, and find that the non-smoothness of the relaxed gradient approximation of linear relaxations negatively affects their optimization. Detailed settings are presented in Appendix A.

### 4.1  Loss Landscape of Certifiable Training

We empirically show that CROWN-IBP [46] and CAP [39] tend to have non-smooth loss landscapes, which hinder optimization during training. We examine the learning curves of IBP and these methods. For a simple analysis on the relation between their optimization and the linear relaxation they used, we avoid considering the mixture of the two logits in (3), and use $\beta = 1$ to consider the CROWN-IBP logit only. Later, we will discuss how $\beta$-scheduling affects the optimization. Figure 1 shows the learning curves on CIFAR-10 under $\epsilon_{\text{train}} = 8/255$. We use $\epsilon_t$-scheduling with the warm-up (regular training, $\epsilon_t = 0$) for the first 10 epochs and the ramp-up ($0 \leq \epsilon_t \leq \epsilon_{\text{train}}$) during epochs 10-130 where we linearly increase the perturbation radius $\epsilon_t$ at iteration $t$ from 0 to the target perturbation $\epsilon_{\text{train}}$. Thus, the training loss may increase even during learning.

In the early phase of the ramp-up period, in which the models are trained with small $\epsilon_t$, CAP and CROWN-IBP have lower losses than IBP as expected because they use much tighter relaxation bounds than IBP. This is consistent with the known results, that CAP tends to outperform the others at small perturbations, such as $\epsilon_{\text{train}} = 2/255$ on CIFAR-10 (see Table 1 for details). However, at the end of the training, when the perturbation reaches its maximum target value ($\epsilon_{\text{train}} = 8/255$), the opposite result is observed where CAP and CROWN-IBP ($\beta = 1$) perform worse than IBP.

To understand this inconsistency, we measure the variation in the loss value along the gradient direction (zeroth-order smoothness) as in Santurkar et al. [30], which is represented as the shaded region in Figure 1. We find that linear relaxation-based methods have large variations, while IBP maintains a small variation throughout the entire training phase. It is known that a smooth loss

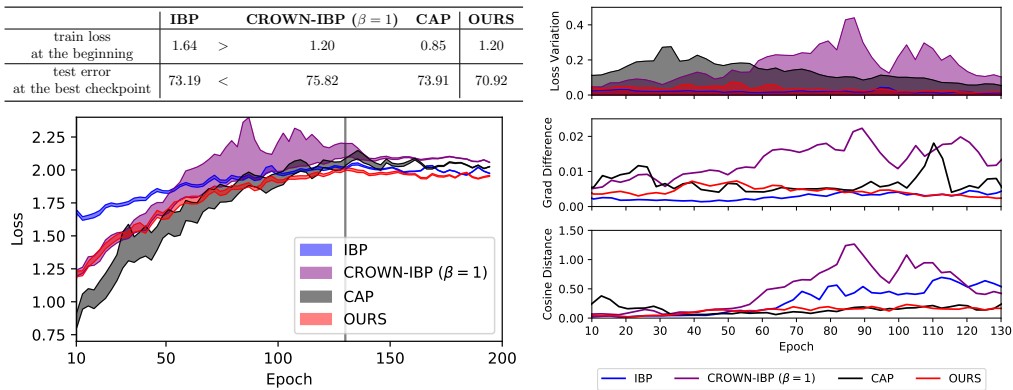

Figure 1: **Left: The learning curves for the scheduled value of $\epsilon_t$ with the loss variation along the gradient descent direction.** The vertical line indicates when the ramp-up ends. IBP starts with a higher loss but ends with a relatively lower loss (error), demonstrating smaller loss variations. Our method uses tight bounds like CROWN-IBP, while its landscape is as favorable as IBP, achieving the best performance among these four methods. We also provide train loss at the beginning and test error at the best checkpoint for the certifiable training methods in the top table. **Right: Non-smoothness measures of the loss landscape between the certifiable training methods.** Lower is better. (Top) Loss variations along the gradient descent direction, (Middle) $\ell_2$-distance between two consecutive loss gradients and (Bottom) the cosine distance between them during the ramp-up phase. Note that they are evaluated with the scheduled value of $\epsilon_t$.

landscape with a small loss variation induces stable and fast optimization with well-behaved gradients [30], which will be discussed in detail in the following section. Therefore, even though CAP and CROWN-IBP ($\beta = 1$) show better robustness in the early phase of training, the non-smooth loss landscape in the ramp-up period hinders the optimization, yielding less robust models.

**Non-smoothness measures of loss landscape** Next, we further investigate the smoothness of the loss landscape to establish a relationship between the optimization behavior and linear relaxation. Figure 1 (Right, middle and bottom) shows the difference between two successive loss gradient steps during training in terms of $\ell_2$- and cosine distance, respectively. We observe that some linear relaxation methods have less smooth loss landscapes with unstable gradients, especially in the early stage of CAP and the middle stage of CROWN-IBP ($\beta = 1$). Moreover, since the gradient directions are not well-aligned, they may not enjoy the advantages of momentum-based optimizers and be sensitive to the learning rate. As will be discussed in the following section, we find that the loss landscape is highly related to the relaxed gradient approximation $\boldsymbol{g}$ used in linear relaxation.

### 4.2 Smoothness of Relaxed Gradient Approximation

In this section, we investigate the optimization behavior further from a theoretical perspective to answer the question: "What makes some loss landscapes more favorable than others?" We find that the relaxed gradient approximation of a linear relaxation affects the smoothness of the landscape. Before that, in the following theorem, we first look at the relation between the optimization and the smoothness of the loss landscape in terms of the Hessian of the loss function with respect to weight parameters. We defer the proof to Appendix F.

**Theorem 1.** *With gradient descent using a step size within an interval $I_t$ during the ramp-up period ($0 \leq \epsilon_t \leq \epsilon$), the loss $\mathcal{L}^\epsilon$ for the target perturbation $\epsilon$ is reduced with*

$$\mathcal{L}^\epsilon(\boldsymbol{\theta}_{t+1}) \leq \mathcal{L}^\epsilon(\boldsymbol{\theta}_t)\big(1 - \frac{\mu}{2}\cos^2(\phi_t)\|\boldsymbol{H}_t^\epsilon \boldsymbol{u}_t\|^{-1}\big) \tag{4}$$

*for $\boldsymbol{u}_t = \frac{\nabla_{\boldsymbol{\theta}}\mathcal{L}^{\epsilon_t}(\boldsymbol{\theta}_t)}{\|\nabla_{\boldsymbol{\theta}}\mathcal{L}^{\epsilon_t}(\boldsymbol{\theta}_t)\|}$ where $0 < \mu \leq \frac{\|\nabla_{\boldsymbol{\theta}}\mathcal{L}^\epsilon\|^2}{2\mathcal{L}^\epsilon}$, $\cos(\phi_t) = \frac{\nabla_{\boldsymbol{\theta}}\mathcal{L}^{\epsilon\,T}\nabla_{\boldsymbol{\theta}}\mathcal{L}^{\epsilon_t}}{\|\nabla_{\boldsymbol{\theta}}\mathcal{L}^\epsilon\|\|\nabla_{\boldsymbol{\theta}}\mathcal{L}^{\epsilon_t}\|}$ and $\boldsymbol{H}_t^\epsilon$ satisfies $\mathcal{L}^\epsilon(\boldsymbol{\theta}_{t+1}) = \mathcal{L}^\epsilon(\boldsymbol{\theta}_t) + \nabla_{\boldsymbol{\theta}}\mathcal{L}^\epsilon(\boldsymbol{\theta}_t)^T\Delta_t + \frac{1}{2}\Delta_t^T\boldsymbol{H}_t^\epsilon\Delta_t$ and $\Delta_t^T\boldsymbol{H}_t^\epsilon\Delta_t > 0$ with $\Delta_t = \boldsymbol{\theta}_{t+1} - \boldsymbol{\theta}_t$.*

Here, $\mu$ is a modified Polyak-Lojasiewicz (PL) constant which is important for the optimization together with the Hessian term $\|\boldsymbol{H}_t^\epsilon \boldsymbol{u}_t\|$ [3, 21]. In this paper, we mainly focus on the Hessian

term. Note that $\boldsymbol{H}_t^\epsilon = \nabla_{\boldsymbol{\theta}}^2 \mathcal{L}^\epsilon(\tilde{\boldsymbol{\theta}}_t)$ is the Hessian of $\mathcal{L}^\epsilon$ at some $\tilde{\boldsymbol{\theta}}_t$ between $\boldsymbol{\theta}_t$ and $\boldsymbol{\theta}_{t+1}$. Thus, since $\Delta_t = -\eta_t \nabla_{\boldsymbol{\theta}} \mathcal{L}^{\epsilon_t}(\boldsymbol{\theta}_t)$ with the learning rate $\eta_t$, we can approximate

$$\|\boldsymbol{H}_t^\epsilon \boldsymbol{u}_t\| \propto \|\boldsymbol{H}_t^\epsilon \Delta_t\| \approx \|\nabla_{\boldsymbol{\theta}} \mathcal{L}^\epsilon(\boldsymbol{\theta}_{t+1}) - \nabla_{\boldsymbol{\theta}} \mathcal{L}^\epsilon(\boldsymbol{\theta}_t)\|. \tag{5}$$

Therefore, it optimizes better with a smaller loss gradient difference (first-order smoothness).

To see what factors influence the loss gradient difference, we first need some mild smoothness assumptions that are natural when the network parameters $\boldsymbol{\theta}_1$ and $\boldsymbol{\theta}_2$ are close to each other, especially they are two consecutive parameters from gradient descent update, i.e., $\boldsymbol{\theta}_t$ and $\boldsymbol{\theta}_{t+1}$.

**Assumption 1.** *Given a linear relaxation method, we make the following assumptions on the bias* $b(\boldsymbol{x}; \boldsymbol{\theta})$ *in the linear relaxation (2) and the probability function* $\boldsymbol{p}(\boldsymbol{x}; \boldsymbol{\theta})$:
*(i)* $\|\nabla_{\boldsymbol{\theta}} b(\boldsymbol{x}; \boldsymbol{\theta}_2) - \nabla_{\boldsymbol{\theta}} b(\boldsymbol{x}; \boldsymbol{\theta}_1)\| \leq L_{\boldsymbol{\theta}\boldsymbol{\theta}}^b \|\boldsymbol{\theta}_2 - \boldsymbol{\theta}_1\|$ *and*
*(ii)* $\|\boldsymbol{p}(\boldsymbol{x}; \boldsymbol{\theta}_2) - \boldsymbol{p}(\boldsymbol{x}; \boldsymbol{\theta}_1)\| \leq L_{\boldsymbol{\theta}}^p \|\boldsymbol{\theta}_2 - \boldsymbol{\theta}_1\|, \forall \boldsymbol{\theta}_1, \boldsymbol{\theta}_2$ *and* $\boldsymbol{x}$.

With the above assumptions, we can provide an upper bound on the loss gradient difference for linear relaxation-based methods to understand the optimization behavior as follows:

**Theorem 2.** *Suppose* $\boldsymbol{x} \in \mathcal{X}$ *is bounded* $\|\boldsymbol{x}\| \leq M$ *with some* $M > 0$. *For a linear relaxation-based method with the upper bound* $\overline{\boldsymbol{s}}_m(\boldsymbol{x}; \boldsymbol{\theta}) = \max_{\boldsymbol{x}' \in \mathbb{B}(\boldsymbol{x}, \epsilon)} \boldsymbol{g}_{\boldsymbol{\theta}}^{(m)}(\boldsymbol{x})^T \boldsymbol{x}' + b_{\boldsymbol{\theta}}^{(m)}(\boldsymbol{x})$, *if each* $b_{\boldsymbol{\theta}}^{(m)}$ *and* $\boldsymbol{p_s}$ *satisfies Assumption 1, then*

$$\|\nabla_{\boldsymbol{\theta}} \mathcal{L}^\epsilon(\boldsymbol{\theta}_2) - \nabla_{\boldsymbol{\theta}} \mathcal{L}^\epsilon(\boldsymbol{\theta}_1)\|$$
$$\leq \mathbb{E}_{(\boldsymbol{x}, y)} \left[ \max_m \left( 2\epsilon \|\nabla_{\boldsymbol{\theta}} \boldsymbol{g}_{\boldsymbol{\theta}_{1,2}}^{(m)}(\boldsymbol{x})\| + M \|\nabla_{\boldsymbol{\theta}} \boldsymbol{g}_{\boldsymbol{\theta}_2}^{(m)}(\boldsymbol{x}) - \nabla_{\boldsymbol{\theta}} \boldsymbol{g}_{\boldsymbol{\theta}_1}^{(m)}(\boldsymbol{x})\| + L^{(m)} \|\boldsymbol{\theta}_2 - \boldsymbol{\theta}_1\| \right) \right] \tag{6}$$

*for any* $\boldsymbol{\theta}_1, \boldsymbol{\theta}_2$, *where* $L^{(m)} = L_{\boldsymbol{\theta}\boldsymbol{\theta}}^{b^{(m)}} + L_{\boldsymbol{\theta}}^{\boldsymbol{p_s}} \|\nabla_{\boldsymbol{\theta}} \overline{\boldsymbol{s}}(\boldsymbol{x}; \boldsymbol{\theta}_{1,2})\|$ *and* $\boldsymbol{\theta}_{1,2}$ *can be any of* $\boldsymbol{\theta}_1$ *and* $\boldsymbol{\theta}_2$.

According to Theorem 2, the relaxed gradient approximations $\boldsymbol{g}^{(m)}$ in the linear relaxation play a major role in shaping the loss landscape. The first two terms in the right hand side of (6) indicate the zeroth- and first-order smoothness of the relaxed gradient approximation with respect to weight parameters, respectively. The smoother the relaxed gradient approximations are, the smoother the loss landscape is. Especially for IBP, using the zero-slope relaxed gradient approximation $\boldsymbol{g}^{(m)} \equiv \boldsymbol{0}$ for all $m$, the loss gradient difference is upper bounded by only the last term, $\max L^{(m)} \|\boldsymbol{\theta}_2 - \boldsymbol{\theta}_1\|$, and it is relatively small for a single gradient step. On the other hand, for other linear relaxation-based methods using non-zero relaxed gradient approximation $\boldsymbol{g}^{(m)} \neq \boldsymbol{0}$, the gradient updates used in the training are more unstable than IBP. It is consistent with the empirical results shown in Figure 1 that there are significant differences between the loss landscape of IBP and others.

## 5 Proposed Method

Our analyses so far suggest that not only tightness of the upper bound on the worst-case loss but also smoothness of the loss landscape is important for building a certifiably robust model. Therefore, we aim to design a new certifiable training method to improve the aforementioned factors (favorable landscape and tighter bound).

**Favorable landscape with smooth $g$ via sparse $\underline{a}$**  To have a smooth loss landscape, we aim to build a linear relaxation with a smooth relaxed gradient approximation by using sparse $\underline{a}$. For example, in a simple 2-layered network case with $\boldsymbol{s}_m(\boldsymbol{x}) = \boldsymbol{w}^T h(\boldsymbol{W}\boldsymbol{x} + \boldsymbol{b}) + b$, we have $\boldsymbol{g}^{(m)} = \boldsymbol{W}^T(\underline{\boldsymbol{a}} \odot \boldsymbol{w}^- + \overline{\boldsymbol{a}} \odot \boldsymbol{w}^+)$ and $\|\nabla_{\boldsymbol{w}} \boldsymbol{g}^{(m)}\| = \|\boldsymbol{W}^T \text{diag}(\underline{\boldsymbol{a}} \odot \mathbf{1}[\boldsymbol{w} < 0] + \overline{\boldsymbol{a}} \odot \mathbf{1}[\boldsymbol{w} \geq 0])\| \leq \|\boldsymbol{W}^T\| (\|\underline{\boldsymbol{a}}\| + \|\overline{\boldsymbol{a}}\|)$ where $\text{diag}(\boldsymbol{v})$ is the diagonal matrix whose entries are the elements of $\boldsymbol{v}$. This simple example implies that a sparse $\underline{\boldsymbol{a}}$ may help to smooth the relaxed gradient approximation.

To this end, we investigate variants of CROWN-IBP with different $\underline{\boldsymbol{a}}$ settings for unstable ReLUs to see their effects on the smoothness of the loss landscape. For each setting, we sample $\underline{a} \in \{0, 1\}$ with different $(p, q)$ with $P(\underline{a} = 1 \mid |l| > |u|) = p$ and $P(\underline{a} = 1 \mid |l| \leq |u|) = q$ for each neuron with pre-activation bounds $l$ and $u$. For example, CROWN-IBP uses $(p, q) = (0, 1)$. We use $\underline{\boldsymbol{a}} = \mathbf{1}[\boldsymbol{u}^+ + \boldsymbol{l}^- > \boldsymbol{0}]$ for the other stable ReLUs. We consider the variants $(p, q) = (0, 0), (0, 0.5), (0, 1), (0.5, 1)$, and $(1, 1)$, in order of decreasing the sparsity. For the other elements of the linear relaxation variable $\phi = \{(\overline{\boldsymbol{a}}, \underline{\boldsymbol{a}}, \overline{\boldsymbol{b}}, \underline{\boldsymbol{b}})\}$, we fix $\overline{\boldsymbol{a}} = \frac{\boldsymbol{u}^+}{\boldsymbol{u}^+ - \boldsymbol{l}^-}, \overline{\boldsymbol{b}} = -\frac{\boldsymbol{u}^+ \boldsymbol{l}^-}{\boldsymbol{u}^+ - \boldsymbol{l}^-}$, and $\underline{\boldsymbol{b}} = \boldsymbol{0}$ for each activation node, because

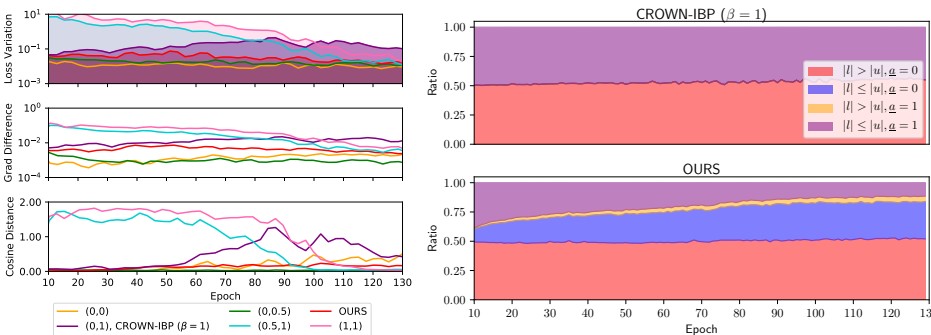

Figure 2: **Left: The non-smoothness measures of the models with different levels of sparsity of $\underline{a}$ during the ramp-up period (same as in Figure 1 (Right)).** Notation $(p, q)$ denotes the variant with sampling $\underline{a} \in \{0, 1\}$ with $P(\underline{a} = 1 \mid |l| > |u|) = p$ and $P(\underline{a} = 1 \mid |l| \le |u|) = q$ for unstable ReLUs. As $\underline{a}$ becomes sparse, the loss landscape is much smoother. **Right: The configuration of $\underline{a}$ used for unstable ReLUs during the ramp-up period.** It indicates that the proposed method reduces the number of nonzero $\underline{a}$ (purple+yellow).

they are the optimal choices for tightening the bound (see Appendix C.2 for details). As expected, Figure 2 (Left) shows that it tends to have a more smooth landscape as $\underline{a}$ becomes more sparse.

However, the sparsity only is not enough to achieve robustness unless the tightness is guaranteed. A variant of CROWN-IBP with $(p, q) = (0, 0)$ achieves a favorable landscape, but they show looser upper bounds which lead to a worse performance (details are presented in Appendix E). Therefore, it is required to search for appropriate $\underline{a}$ that can achieve both tightness and favorable landscape.

**Tighter bound via optimization**   Now, we aim to make $\underline{a}$ sparse and to tighten the upper bound in (1), simultaneously. We can achieve both by minimizing the upper bound over the linear relaxation variable $\phi$ as follows: $\mathcal{L}(\overline{s}(x)) \ge \min_\phi \mathcal{L}(\overline{s}(x; \phi)) \ge \max_{x' \in \mathbb{B}(x, \epsilon)} \mathcal{L}(f(x'))$. It can be equivalently understood as solving the dual optimization in CAP rather than using a dual feasible solution. However, solving the dual optimization is computationally prohibited for the linear relaxation of CAP. To resolve this problem, we use the same linear relaxation as IBP for the subnetworks of $s$ except for $s$ itself, similar to CROWN-IBP. Further, we efficiently compute a surrogate $\hat{\underline{a}}$ of the minimizer $\underline{a}^* = \arg\min_{\underline{a}} \mathcal{L}(\overline{s}(x; \phi = \{(\overline{a}, \underline{a}, \overline{b}, \underline{b})\}))$ using the one-step projected gradient update of the relaxation variable $\underline{a}$ for unstable ReLUs. Specifically, we have

$$\hat{\underline{a}} = \Pi_{[0,1]^n} \left( \underline{a}_0 - \eta \text{sign}(\nabla_{\underline{a}} \mathcal{L}(\overline{s}(x; \phi_0))) \right) \tag{7}$$

with an initial variable $\phi_0 = \{(\overline{a}, \underline{a}_0, \overline{b}, \underline{b})\}$ where $\underline{a}_0 \sim U[0, 1]^n$ and $\eta \ge 1$, yielding the final objective $\mathcal{L}(\overline{s}(x; \hat{\phi}))$ where $\hat{\phi} = \{(\overline{a}, \hat{\underline{a}}, \overline{b}, \underline{b})\}$.

The update (7) *implicitly* leads to a sparse $\hat{\underline{a}}$ since the update direction tends to be negative for unstable ReLUs. We get the motivation from the relaxation of CROWN-IBP explained in Appendix C.1. By tracking the sign of each element in computing the update direction, we can conclude that (7) yields a sparse $\hat{\underline{a}}$. When computing the upper bound $\overline{s}$, the lower slope $\underline{a} \in [0, 1]^n$ is multiplied with non-positive $w^- \le 0$, which yields $w_j^- \underline{a}_j \le 0$, and thus the preactivation value $z_j$ is likely to be (close to) the lower bound $l_j$ to maximize the value $\sum_{j:w_j<0} w_j \underline{a}_j z_j$ for the element $j$ s.t. $w_j < 0$. Thus, $\nabla_{\underline{a}_j} \overline{s}_m(x) = w_j^- l_j \ge 0$. Thus, when projected onto $[0, 1]^n$, the update direction, $-\eta \text{sign}(\nabla_{\underline{a}} \mathcal{L}(\overline{s}(x; \phi_0), y)) = -\eta \text{sign}(\sum_{m \ne y} p_m \nabla_{\underline{a}} \overline{s}_m)$, is likely to yield a sparser $\hat{\underline{a}}$ than CROWN-IBP (cf. Appendix F). Similar logic applies to the inner layers.

## 6   Experiments

In this section, we demonstrate the proposed method satisfies two key criteria required for building certifiably robust models: 1) tightness of the upper bound on the worst-case loss, and 2) smoothness of the loss landscape. Subsequently, we evaluate the performance of the method by comparing with others certifiable training methods. Details on the experimental settings are in Appendix A.

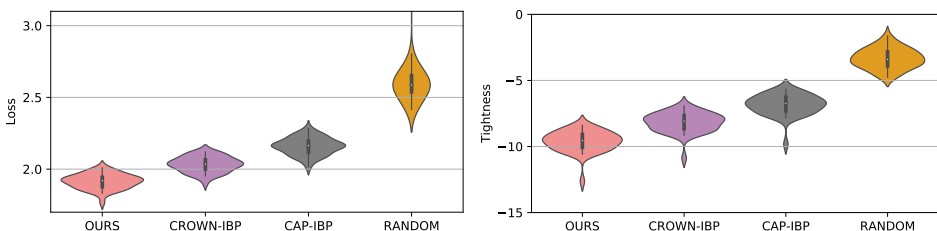

Figure 3: **Violin plots of the test loss (*Left*) and of tightness (*Right*) for various linear relaxations.** Lower is better. This shows that the proposed relaxation method has a tighter bound than the others.

Table 1: Test errors (Standard / PGD / Verified error) of IBP, CROWN-IBP ($\beta = 1$), CAP, and OURS. Bold and underline numbers are the first and second lowest verified error. We use two different models (Shallow/Deep) for CIFAR-10 detailed in Appendix A.

| Data | $\epsilon_{\text{test}}(l_\infty)$ | IBP | CROWN-IBP ($\beta = 1$) | CAP | OURS |
|---|---|---|---|---|---|
| **MNIST** | 0.1 | 1.18 / 2.16 / 3.52 | 1.07 / 1.69 / **2.10** | 0.80 / 1.73 / 3.19 | 1.09 / 1.77 / 2.36 |
| | 0.2 | 2.00 / 3.29 / 6.31 | 2.99 / 5.50 / 7.97 | 3.22 / 6.72 / 11.06 | 1.70 / 3.44 / **4.34** |
| | 0.3 | 3.50 / 5.85 / 10.45 | 5.73 / 10.76 / 16.28 | 19.19 / 35.84 / 47.85 | 3.49 / 5.59 / **9.79** |
| | 0.4 | 3.50 / 7.30 / 17.96 | 5.73 / 14.63 / 23.80 | - | 3.49 / 6.77 / **15.42** |
| **CIFAR-10 (Shallow)** | $2/255$ | 37.98 / 49.40 / 55.39 | 32.48 / 42.77 / 50.15 | 28.80 / 38.95 / **48.50** | 31.49 / 42.73 / 49.42 |
| | $4/255$ | 46.42 / 57.42 / 62.80 | 45.56 / 58.24 / 64.47 | 40.78 / 52.62 / 61.88 | 42.53 / 55.55 / **61.52** |
| | $6/255$ | 52.84 / 63.92 / 68.79 | 54.72 / 65.28 / 71.04 | 49.20 / 60.85 / 69.03 | 50.19 / 61.88 / **66.90** |
| | $8/255$ | 55.71 / 66.79 / 70.95 | 61.37 / 70.66 / 75.37 | 56.77 / 66.78 / 73.02 | 56.01 / 66.17 / **69.70** |
| | $16/255$ | 67.10 / 75.12 / 78.26 | 76.65 / 81.90 / 84.42 | 75.11 / 80.67 / 82.07 | 65.93 / 75.39 / **77.87** |
| **CIFAR-10 (Deep)** | $2/255$ | 39.17 / 48.80 / 55.48 | 29.02 / 40.17 / **46.22** | - | 31.48 / 42.52 / 47.89 |
| | $8/255$ | 59.53 / 65.98 / 70.86 | 59.43 / 65.79 / 73.34 | - | 50.78 / 62.58 / **68.44** |
| **SVHN** | 0.01 | 19.91 / 34.12 / 43.83 | 17.25 / 30.84 / 39.88 | 16.88 / 30.16 / **37.09** | 16.41 / 30.43 / 39.44 |

**Tightness** To validate that the proposed method (OURS) has sufficiently tight bounds, we analyze various linear relaxation methods in Figure 3. We define a tightness measure as $\sum_{m=0}^{c-1} \overline{s}_m(\boldsymbol{x})$. Then, we evaluate multiple methods on a single fixed model pre-trained with the proposed training method. The compared methods are, from left to right, OURS, CROWN-IBP, CAP-IBP, and RANDOM. All methods use the same IBP relaxation for subnetworks, but use different linear relaxation variables $\underline{\boldsymbol{a}}$ for the whole network $\boldsymbol{s}$. CROWN-IBP, CAP-IBP, and RANDOM use $\underline{\boldsymbol{a}} = \mathbf{1}[\boldsymbol{u}^+ + \boldsymbol{l}^- > 0]$, $\underline{\boldsymbol{a}} = \frac{\boldsymbol{u}^+}{\boldsymbol{u}^+ - \boldsymbol{l}^-}$ and $\underline{\boldsymbol{a}} \sim U[0,1]^n$, respectively. We fix the other variables $\overline{\boldsymbol{a}}, \overline{\boldsymbol{b}}$, and $\underline{\boldsymbol{b}}$, as in Section 5. In both figures, our method shows the lowest value on average, which indicates that a single gradient step in (7) is sufficient to obtain tighter bounds. See Appendix M for other models.

**Smoothness** Figure 2 (Right) shows that our method has successfully yielded more sparse $\underline{\boldsymbol{a}}$ than CROWN-IBP ($\beta = 1$). This leads to the results shown in Figure 1 (Right) that the proposed method has a smooth loss landscape as with IBP, whereas the others have less smooth ones.

**Certified Robustness** We evaluate the performance of the proposed method and compare it to other certifiable training methods: IBP [15], CROWN-IBP ($\beta = 1$) [46], and CAP [40], as in Section 4.1. On MNIST, we follow Zhang et al. [46] and use $\epsilon_{\text{train}} \geq \epsilon_{\text{test}}$; whereas for CAP, we use the same $\epsilon_{\text{train}} = \epsilon_{\text{test}}$ which yields better results. We used three evaluation metrics: standard (clean), 100-step PGD, and verified error. For the verified error, we evaluated with the bound $\overline{s}$ of each method.

Table 1 summarizes the evaluation results under different $\epsilon_{\text{test}}$ for each dataset. In general, when $\epsilon_{\text{test}}$ is low, methods with tighter linear relaxations show good performance, whereas IBP tends to perform better as $\epsilon_{\text{test}}$ increases. In short, the current SOTA methods perform well for a specific range of $\epsilon_{\text{test}}$. For example, IBP shows relatively better performance in the case of $\epsilon_{\text{test}} = 0.3, 0.4$ on MNIST and $\epsilon_{\text{test}} = 6/255, 8/255, 16/255$ on CIFAR-10. On the other hand, CAP and CROWN-IBP ($\beta = 1$) outperform IBP in the case of $\epsilon_{\text{test}} = 0.1$ on MNIST, $\epsilon_{\text{test}} = 2/255$ on CIFAR-10 and $\epsilon_{\text{test}} = 0.001$ on SVHN. This result is consistent with the analysis shown in Figure 1 that CAP and CROWN-IBP ($\beta = 1$) have lower loss than IBP at small $\epsilon$, but their loss landscape is less smooth than IBP, leading

Table 2: Test errors (Standard / Verified error) compared to the best errors reported in the literature. Bold numbers are the lowest verified error. The results in RS [42] and COLT [2] are evaluated with a MILP based exact verifier [36].

| Data | MNIST | | CIFAR-10 | |
|---|---|---|---|---|
| $\epsilon_{\text{test}}(l_\infty)$ | 0.1 | 0.3 | $2/255$ | $8/255$ |
| RS | 1.32 / 4.87 | 2.67 / 19.32 | 38.88 / 54.07 | 59.55 / 79.72 |
| DiffAI | 1.3 / 4.2 | 3.4 / 10.7 | 37.7 / 54.5 | 53.8 / 72.8 |
| COLT | 0.8 / 2.9 | 2.7 / 14.3 | 21.6 / **39.5** | 48.3 / 72.5 |
| OURS | 1.09 / **2.28** | 2.42 / **7.84** | 31.48 / 47.89 | 50.78 / **68.44** |

to worse performance at large $\epsilon$. Moreover, CAP cannot be trained on MNIST when $\epsilon_{\text{train}} = 0.4$. As the case is also not specified in Wong et al. [40], it seems that CAP is hard to be robust to $\epsilon_{\text{train}} \geq 0.4$. On the other hand, the proposed method shows consistent performance in a wide range of $\epsilon_{\text{test}}$ values, achieving the best performance in most cases, since it has tighter bounds and a favorable landscape, not overfitting to a local minimum during the $\epsilon_t$-scheduling. We also conduct additional experiments on the hyperparameters in Appendix I, J, and K.

We also compared our method with other prior work, ReLU Stability (RS) [42], DiffAI [24] and COLT [2], in Table 2. All experiments and results, except for Table 2, in this paper are based on our own reimplementation. The proposed method shows the best verified error, except for the case $\epsilon_{\text{test}} = 2/255$ on CIFAR-10, in which COLT performs better. This is because COLT is based on the linear relaxation used in CAP.

Unlike standard training, certifiable training requires $\epsilon_t$-scheduling. It is implicitly assumed a set of weights that makes the network robust to a small $\epsilon_t$ is a good initial point to learn robustness to a large $\epsilon_{\text{train}}$. However, training methods with tighter bounds start with a lower loss at a small $\epsilon_t$, but with an unfavorable loss landscape, they cannot explore a sufficiently large area of the parameter space. Hence, they overfit to be robust to a small perturbation, and cannot generalize to a large perturbation. CAP and CROWN-IBP ($\beta = 1$) are typical examples that demonstrate the overfitting. This may overegularize the weight norm and decrease the model capacity [40, 46]. The tightness of the proposed method improves the performance for a small $\epsilon_{\text{train}}$, while the smoothness helps the optimization process, which also leads to better performance for a large $\epsilon_{\text{train}}$. To conclude, the proposed method can achieve a decent performance under a wide range of perturbations.

## 7   Discussion

A proper $\beta$-scheduling in (8) could help the model to obtain a smoother landscape and to improve the robustness performance. We use two different settings of $\beta$, $\text{CBP}_{1\to1}$ and $\text{CBP}_{1\to0}$, where CBP stands for CROWN-IBP and the subscript $\beta_{\text{start}} \to \beta_{\text{end}}$ refers to the linear scheduling on $\beta$ from $\beta_{\text{start}}$ to $\beta_{\text{end}}$. Zhang et al. [46] found that the $\beta$-scheduling of $\text{CBP}_{1\to0}$ can improve the robustness performance. And they argued that this is because training with a tighter bound of CROWN-IBP at the beginning can provide a good initialization for later IBP training. On the other hand, we provide another explanation that $\text{CBP}_{1\to0}$ starts with a tighter bound (CROWN-IBP only) but not overfits to small perturbation by gradually introducing the IBP objective which has a smoother landscape. In Appendix P, we show that the success of $\text{CBP}_{1\to0}$ is due to the smoothness.

In the case of $\epsilon_{\text{test}} = 8/255$, $\text{CBP}_{1\to0}$ obtains the improved standard/PGD/verified error of 54.02/65.42/**66.94** in the literature [46] while $\text{CBP}_{1\to1}$ shows 59.43/65.79/73.34. The result also implies that the smoothness is a crucial factor in certifiable training, especially in a large $\epsilon_{\text{train}}$. The proposed method achieves **50.78/62.58**/68.44. Although we couldn't achieve the same verification performance ($\Delta = -1.50$), it has better standard/PGD accuracy by a larger margin ($\Delta = +3.24/ + 2.84$). For $\epsilon_{\text{test}} = 2/255$, however, $\text{CBP}_{1\to0}$ shows 35.83/46.21/52.26, worse than 29.02/40.17/46.22 of $\text{CBP}_{1\to1}$. For a small $\epsilon_{\text{test}}$, the tightness affects the performance significantly. The proposed method shows 31.48/42.52/47.89, the similar performance using a tighter bound ($\text{CBP}_{1\to1}$).

To summarize, we emphasize two points in this section: (1) the $\beta$-scheduling in $\text{CBP}_{1\to0}$ improves the smoothness by introducing the IBP objective, and (2) while the two CROWN-IBP methods perform well only for a specific range of perturbations due to lack of either the tightness or the smoothness,

the proposed method has the two desired properties even without introducing the mixture parameter $\beta$ and it shows a decent performance in a wide range of adversarial perturbations.

## Acknowledgement

Sungyoon Lee is supported by a KIAS Individual Grant (AP083601) via the Center for AI and Natural Sciences at Korea Institute for Advanced Study. This work was supported by the National Research Foundation of Korea (NRF) grant funded by the Korean government (MSIT) (NRF-2019R1A2C2002358).

## 8 Conclusion

We have investigated the loss landscape of certifiable training and found that the smoothness of the loss landscape is an important factor that influences in building certifiably robust models. To this end, we proposed a method that satisfies the two criteria: tightness of the upper bound on the worst-case loss and smoothness of the loss landscape. With the two factors, the proposed method can achieve a decent performance for a wide range of perturbations, while others with only one of the two factors can perform well only for a specific range of perturbations. We believe that with a better understanding of the loss landscape, better certifiably robust models can be built.

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
