# A Experimental Settings

**Datasets and Architectures** In the experiments, we use three datasets: MNIST, CIFAR-10 and SVHN and model architectures (Small, Medium, and Large) in Gowal et al. [15] and their variants (Small* and Large*) as follows:

- Small: Conv(·,16,4,2) - Conv(16,32,4,1) - Flatten - FC(·,100) - FC(100,c)
- Small*: Conv(·,16,4,2) - Conv(16,32,4,2) - Flatten - FC(·,100) - FC(100,c)
- Medium: Conv(·,32,3,1) - Conv(32,32,4,2) - Conv(32,64,3,1) - Conv(64,64,4,2) - Flatten - FC(·,512) - FC(512,512) - FC(512,c)
- Large: Conv(·,64,3,1) - Conv(64,64,3,1) - Conv(64,128,3,2) - Conv(128,128,3,1) - Conv(128,128,3,1) - Flatten - FC(·,512) - FC(512,c)
- Large*: Conv(·,64,3,1) - Conv(64,128,3,2) - Conv(128,128,3,1) - Conv(128,128,3,1) - Flatten - FC(·,512) - FC(512,c)

where $\text{Conv}(c_1, c_2, k, s)$ is a conv layer with input channel $c_1$, output channel $c_2$, kerner size $k$, and stride $s$, and $\text{FC}(d_1, d_2)$ is a fully-connected layer with input dimension $d_1$ and output dimension $d_2$. All layers are followed by ReLU activation except for the last layer and the flatten layer (Flatten).

**Loss and training schedules** For general training schedules, we refer to Appendix C, D of Zhang et al. [46] with a single GPU (Titan Xp). We use the following mixed cross-entropy loss as in Zhang et al. [46]:

$$\kappa \mathcal{L}\left(f(\boldsymbol{x}; \boldsymbol{\theta})\right) + (1-\kappa)\mathcal{L}\left((1-\beta)\overline{s}^{\text{IBP}}(\boldsymbol{x}; \boldsymbol{\theta}) + \beta \overline{s}^{\text{MODEL}}(\boldsymbol{x}; \boldsymbol{\theta})\right), \tag{8}$$

where $\kappa$ is the mixing weight between the natural loss and the robust loss, and $\beta$ is the mixing weight between the two bounds obtained with IBP and given relaxation method (e.g. CROWN-IBP).

## A.1 Settings in Section 4.1

**Figure 1 (Left)** We conduct the experiment in Figure 1 on CIFAR-10 dataset with Medium architecture over all four methods. We train the model with $\epsilon_{train} = 8/255$ for 200 epochs using $\epsilon_t$-scheduling with 10 warm-up epochs and 120 ramp-up epochs. We use Adam optimizer with learning rate 0.001. We reduce the learning rate by 50% every 10 epochs after $\epsilon_t$-scheduling ends.

To demonstrate the instability of each training, we describe the variation of the loss along the gradient direction as Santurkar et al. [30]. We take steps of different lengths in the direction of the gradient and measure the loss values obtained at each step. For the sake of consistency, we fix a Cauchy random matrix when evaluating CAP to obtain deterministic loss landscapes, not introducing randomness. The loss variation is computed with

$$\mathcal{L}^{\epsilon_t}(\boldsymbol{\theta}([0,5])) \text{ where } \boldsymbol{\theta}(\lambda) \equiv \boldsymbol{\theta}_t - \lambda\eta\nabla_{\boldsymbol{\theta}}\mathcal{L}^{\epsilon_t}(\boldsymbol{\theta}_t), \tag{9}$$

where $\boldsymbol{\theta}_t$ is the current model parameters and $\eta$ is the learning rate. For the step of length $\lambda$, we sample ten points from a range of [0,5] on a log scale.

**Figure 1 (Right)** We plot the $\ell_2$- and cosine distance between two successive loss gradient steps during training as follows:

$$\text{Grad Difference (Middle)} = \|\nabla_{\boldsymbol{\theta}}\mathcal{L}^{\epsilon_t}(\boldsymbol{\theta}_t) - \nabla_{\boldsymbol{\theta}}\mathcal{L}^{\epsilon_t}(\boldsymbol{\theta}_{t+1})\| \text{ and}$$
$$\text{Cosine Distance (Bottom)} = 1 - \cos(\nabla_{\boldsymbol{\theta}}\mathcal{L}^{\epsilon_t}(\boldsymbol{\theta}_t), \nabla_{\boldsymbol{\theta}}\mathcal{L}^{\epsilon_t}(\boldsymbol{\theta}_{t+1})),$$

where $\cos(\boldsymbol{v}_1, \boldsymbol{v}_2)$ is the cosine value of the angle between two vectors $\boldsymbol{v}_1$ and $\boldsymbol{v}_2$.

## A.2 Settings in Table 1

For MNIST, we use the same hyper-parameters as in Appendix C of Zhang et al. [46]. We train for 200 epochs (10 warm-up epochs and 50 ramp-up epochs) on Large model with batch sizes of 100. we decay the learning rate, 0.0005, by 10% in [130,190] epochs. As mentioned in Zhang et al. [46], we also found the same issue when training with small $\epsilon$ (see Appendix K for details). To alleviate the issue, we use $\epsilon_{train} = \min(0.4, \epsilon_{test} + 0.1)$ for each $\epsilon_{test}$ as Table 2 of Zhang et al. [46].

For CIFAR-10, we train for 400 epochs (20 warm-up epochs and 240 ramp-up epochs) on Medium model with batch sizes of 128. We decay the learning rate, 0.003, by $2\times$ every 10 epochs after the ramp-up period.

For SVHN, we train for 200 epochs (10 warm-up epochs and 120 ramp-up epochs) on Large model with batch sizes of 128 (OURS with batch sizes of 80 to avoid out of memory). We decay the learning rate, 0.0003, by $2\times$ every 10 epochs after the ramp-up period. Only for SVHN, we apply normalization with mean (0.438, 0.444, 0.473) and standard deviation (0.198, 0.201, 0.197) for each channel.

In Table 1, we use $\kappa$-scheduling from 1 to 0. For the corresponding results of $\kappa$-scheduling from 0 to 0, we refer the reader to Table 4.

We modify the source code for CAP[1] to match our settings. For example, we introduce the warm-up period and linear $\epsilon$-scheduling. We avoid using the reported results in the literature and aim to make a fair comparison under the same settings with only minor differences - for example, because CAP does not support the channel-wise normalization, we could not use the input normalization. Also, due to the memory limit of CAP, we use a smaller batch size of 32 and try other smaller architectures. We found that it often achieves better results with smaller architectures (similar to the results in Table 3 of Wong et al. [40]). Thus, we present the performance with Large*, Medium, and Small* on MNIST, CIFAR-10, and SVHN, respectively. Throughout the experiments, CAP uses the fixed $\kappa = 0$.

## B    Interval Bound Propagation (IBP)

IBP [15] starts from the interval bound $\mathcal{I}^{(0)} \equiv \{ \boldsymbol{z} : \boldsymbol{l}^{(0)} \leq \boldsymbol{z} \leq \boldsymbol{u}^{(0)} \} = \mathbb{B}(\boldsymbol{x}, \epsilon)$ in the input space with the upper bound $\boldsymbol{u}^{(0)} = \boldsymbol{x} + \epsilon \boldsymbol{1}$ and the lower bound $\boldsymbol{l}^{(0)} = \boldsymbol{x} - \epsilon \boldsymbol{1}$ where $\boldsymbol{1}$ is a column vector filled with 1. Then we propagate the interval bound $\mathcal{I}^{(k-1)} \equiv \{ \boldsymbol{z} : \boldsymbol{l}^{(k-1)} \leq \boldsymbol{z} \leq \boldsymbol{u}^{(k-1)} \}$ by using following equations iteratively:

$$\boldsymbol{u}^{(k)} = h^{(k)}(\boldsymbol{u}^{(k-1)}) \text{ and } \boldsymbol{l}^{(k)} = h^{(k)}(\boldsymbol{l}^{(k-1)}) \tag{10}$$

for element-wise monotonic increasing nonlinear activation $h^{(k)}$ with the pre-activation bounds $\boldsymbol{u}^{(k-1)}$ and $\boldsymbol{l}^{(k-1)}$, and

$$\boldsymbol{u}^{(k)} = \boldsymbol{W}^{(k)} \left( \frac{\boldsymbol{u}^{(k-1)} + \boldsymbol{l}^{(k-1)}}{2} \right) + |\boldsymbol{W}^{(k)}| \left( \frac{\boldsymbol{u}^{(k-1)} - \boldsymbol{l}^{(k-1)}}{2} \right) + \boldsymbol{b}^{(k)} \text{ and}$$

$$\boldsymbol{l}^{(k)} = \boldsymbol{W}^{(k)} \left( \frac{\boldsymbol{u}^{(k-1)} + \boldsymbol{l}^{(k-1)}}{2} \right) - |\boldsymbol{W}^{(k)}| \left( \frac{\boldsymbol{u}^{(k-1)} - \boldsymbol{l}^{(k-1)}}{2} \right) + \boldsymbol{b}^{(k)}$$

for linear function $h^{(k)}$ ($k = 1, \cdots, K$). Finally, IBP uses the worst-case margin $\bar{\boldsymbol{s}} = \boldsymbol{u}^{(K)}$ to formulate the objective in (1) for certifiable training.

---

[1]https://github.com/locuslab/convex_adversarial

## C  Details on Linear Relaxation

### C.1  Linear relaxation explained in CROWN [45]

To make the paper self-contained, we provide details of linear relaxation given in the supplementary material of CROWN [45]. We refer readers to the supplementary for more details. Given a network $h^{[k]}$, we want to upper bound the activation $h_i^{[k]}$. We have $h_i^{[k]}(\boldsymbol{x}') = \boldsymbol{W}_{i,:}^{(k)} h^{(k-1)}(h^{[k-2]}(\boldsymbol{x}')) + \boldsymbol{b}_i^{(k)} = \boldsymbol{W}_{i,:}^{(k)} h^{(k-1)}(\boldsymbol{z}^{(k-2)'}) + \boldsymbol{b}_i^{(k)}$ where $\boldsymbol{z}^{(k-2)'} = h^{[k-2]}(\boldsymbol{x}')$. With the linear function bounds of $\overline{h}^{(k-1)}$ and $\underline{h}^{(k-1)}$ on the activation function $h^{(k-1)}$, we have

$$
\begin{aligned}
h_i^{[k]}(\boldsymbol{x}') =& \boldsymbol{W}_{i,:}^{(k)} h^{(k-1)}(\boldsymbol{z}^{(k-2)'}) + \boldsymbol{b}_i^{(k)} \\
\leq& \sum_{\boldsymbol{W}_{i,j}^{(k)}<0} \boldsymbol{W}_{i,j}^{(k)} \underline{h}_j^{(k-1)}(\boldsymbol{z}^{(k-2)'}) + \sum_{\boldsymbol{W}_{i,j}^{(k)}\geq 0} \boldsymbol{W}_{i,j}^{(k)} \overline{h}_j^{(k-1)}(\boldsymbol{z}^{(k-2)'}) + \boldsymbol{b}_i^{(k)} \\
=& \sum_{\boldsymbol{W}_{i,j}^{(k)}<0} \boldsymbol{W}_{i,j}^{(k)} \underline{a}_j^{(k-1)} \boldsymbol{z}_j^{(k-2)'} + \sum_{\boldsymbol{W}_{i,j}^{(k)}\geq 0} \boldsymbol{W}_{i,j}^{(k)} \overline{a}_j^{(k-1)} \boldsymbol{z}_j^{(k-2)'} \\
&+ \sum_{\boldsymbol{W}_{i,j}^{(k)}<0} \boldsymbol{W}_{i,j}^{(k)} \underline{b}_j^{(k-1)} + \sum_{\boldsymbol{W}_{i,j}^{(k)}\geq 0} \boldsymbol{W}_{i,j}^{(k)} \overline{b}_j^{(k-1)} + \boldsymbol{b}_i^{(k)} \\
=& \tilde{\boldsymbol{W}}_{i,:}^{(k)} \boldsymbol{z}^{(k-2)'} + \tilde{\boldsymbol{b}}_i^{(k)} \\
=& \tilde{\boldsymbol{W}}_{i,:}^{(k)} h^{[k-2]}(\boldsymbol{x}') + \tilde{\boldsymbol{b}}_i^{(k)} \\
=& \tilde{\boldsymbol{W}}_{i,:}^{(k)} \left( \boldsymbol{W}^{(k-2)}(h^{[k-3]}(\boldsymbol{x}')) + \boldsymbol{b}^{(k-2)} \right) + \tilde{\boldsymbol{b}}_i^{(k)} \\
=& \hat{\boldsymbol{W}}_{i,:}^{(k-2)} h^{(k-3)}(\boldsymbol{z}^{(k-3)'}) + \hat{\boldsymbol{b}}_i^{(k-2)},
\end{aligned}
$$

where $\tilde{\boldsymbol{W}}_{i,:}^{(k)} = \boldsymbol{W}_{i,:}^{(k)} \boldsymbol{D}^{(k-1)}$ with the diagonal matrix $\boldsymbol{D}_{j,j}^{(k-1)} = \underline{a}_j^{(k-1)}$ for $j$ satisfying $\boldsymbol{W}_{i,j}^{(k)} < 0$ and $\boldsymbol{D}_{j,j}^{(k-1)} = \overline{a}_j^{(k-1)}$ for $j$ satisfying $\boldsymbol{W}_{i,j}^{(k)} \geq 0$, $\tilde{\boldsymbol{b}}_i^{(k)} = \sum_{\boldsymbol{W}_{i,j}^{(k)}<0} \boldsymbol{W}_{i,j}^{(k)} \underline{b}_j^{(k-1)} + \sum_{\boldsymbol{W}_{i,j}^{(k)}\geq 0} \boldsymbol{W}_{i,j}^{(k)} \overline{b}_j^{(k-1)} + \boldsymbol{b}_i^{(k)}$, $\hat{\boldsymbol{W}}_{i,:}^{(k-2)} = \tilde{\boldsymbol{W}}_{i,:}^{(k)} \boldsymbol{W}^{(k-2)}$, and $\hat{\boldsymbol{b}}_i^{(k-2)} = \tilde{\boldsymbol{W}}_{i,:}^{(k)} \boldsymbol{b}^{(k-2)} + \tilde{\boldsymbol{b}}_i^{(k)}$. Applying similar method iteratively, we can obtain $g$ and $b$ in (2) for the linear relaxation of $h_i^{[k]}$.

### C.2  Dual Optimization View

We first modify some notations in the main paper and use the notations similar to Wong and Kolter [39]. We use the following hat notations: $\hat{\boldsymbol{z}}^{(k+1)} = \boldsymbol{W}^{(k+1)} \boldsymbol{z}^{(k)} + \boldsymbol{b}^{(k+1)}$ and $\boldsymbol{z}^{(k)} = h^{(k)}(\hat{\boldsymbol{z}}^{(k)})$ where $h^{(k)}$ is the $k$-th nonlinear activation function. We can build a primal problem with $\boldsymbol{c}^T = \boldsymbol{C}_{m,:}$ as follows:

$$
\max_{\boldsymbol{z}^{(K)}} \boldsymbol{c}^T \hat{\boldsymbol{z}}^{(K)} \tag{11}
$$

such that

$$
\begin{aligned}
\boldsymbol{x} - \epsilon \boldsymbol{1} &\leq \boldsymbol{z}^{(0)}, \\
\boldsymbol{z}^{(0)} &\leq \boldsymbol{x} + \epsilon \boldsymbol{1}, \\
\hat{\boldsymbol{z}}^{(k+1)} &= \boldsymbol{W}^{(k+1)} \boldsymbol{z}^{(k)} + \boldsymbol{b}^{(k+1)} \ (k = 0, \cdots, K-1), \text{ and} \\
\boldsymbol{z}^{(k)} &= h^{(k)}(\hat{\boldsymbol{z}}^{(k)}) \ (k = 1, \cdots, K-1).
\end{aligned}
$$

Note that our $c$ is negation of that of Wong and Kolter [39]. Now we can derive the dual of the primal (11) as follows:

$$\min_{\substack{\boldsymbol{\xi}^+,\boldsymbol{\xi}^-\geq 0\\ \boldsymbol{\nu}_k}} \sup_{\boldsymbol{z}^{(k)},\hat{\boldsymbol{z}}^{(k)}} \boldsymbol{c}^T\hat{\boldsymbol{z}}^{(K)} + \boldsymbol{\xi}^{-T}(\boldsymbol{x}-\epsilon\mathbf{1}-\boldsymbol{z}^{(0)}) + \boldsymbol{\xi}^{+T}(\boldsymbol{z}^{(0)}-\boldsymbol{x}-\epsilon\mathbf{1})$$

$$+ \sum_{k=0}^{K-1}\boldsymbol{\nu}_{k+1}^T\left(\hat{\boldsymbol{z}}^{(k+1)}-(\boldsymbol{W}^{(k+1)}\boldsymbol{z}^{(k)}+\boldsymbol{b}^{(k+1)})\right) + \sum_{k=1}^{K-1}\hat{\boldsymbol{\nu}}_k^T\left(\boldsymbol{z}^{(k)}-h^{(k)}(\hat{\boldsymbol{z}}^{(k)})\right)$$

$$= (\boldsymbol{c}+\boldsymbol{\nu}_K)^T\hat{\boldsymbol{z}}^{(K)} + (\boldsymbol{\xi}^+-\boldsymbol{\xi}^--\boldsymbol{W}^{(1)T}\boldsymbol{\nu}_1)^T\boldsymbol{z}^{(0)} + \sum_{k=1}^{K-1}(-\boldsymbol{W}^{(k+1)T}\boldsymbol{\nu}_{k+1}+\hat{\boldsymbol{\nu}}_k)^T\boldsymbol{z}^{(k)}$$

$$+ \sum_{k=1}^{K-1}(\hat{\boldsymbol{\nu}}_k^T h^{(k)}(\hat{\boldsymbol{z}}^{(k)}) - \boldsymbol{\nu}_k^T\hat{\boldsymbol{z}}^{(k)}) \tag{12}$$

$$- \boldsymbol{\nu}_1^T\boldsymbol{b}^{(1)} - \boldsymbol{\xi}^T\boldsymbol{x} - \epsilon\|\boldsymbol{\xi}\|_1.$$

It leads to $\boldsymbol{c}+\boldsymbol{\nu}_K=\mathbf{0}, \boldsymbol{\xi}^+-\boldsymbol{\xi}^--\boldsymbol{W}^{(1)T}\boldsymbol{\nu}_1=\mathbf{0}$, and $-\boldsymbol{W}^{(k+1)T}\boldsymbol{\nu}_{k+1}+\hat{\boldsymbol{\nu}}_k=\mathbf{0}$ $(k=1,\cdots,K-1)$. Alternatively, they are represented as follows:

$$\boldsymbol{\nu}_K = -\boldsymbol{c},$$
$$\hat{\boldsymbol{\nu}}_k = \boldsymbol{W}^{(k+1)T}\boldsymbol{\nu}_{k+1} \ (k=K-1,\cdots,1), \text{ and}$$
$$\boldsymbol{\xi} = \hat{\boldsymbol{\nu}}_1.$$

Now we need relationship between $\hat{\boldsymbol{\nu}}_k$ and $\boldsymbol{\nu}_k$, i.e., $\boldsymbol{\nu}_k = g(\hat{\boldsymbol{\nu}}_k)$. With the further relaxation $\boldsymbol{\nu}_k = \boldsymbol{\alpha}_k \odot \hat{\boldsymbol{\nu}}_k$, we have a relaxed problem as follows:

$$\min_{\boldsymbol{\alpha}_k} \sup_{\boldsymbol{z}^{(k)},\hat{\boldsymbol{z}}^{(k)}} \sum_{k=1}^{K-1}(\hat{\boldsymbol{\nu}}_k^T h^{(k)}(\hat{\boldsymbol{z}}^{(k)}) - \boldsymbol{\nu}_k^T\hat{\boldsymbol{z}}^{(k)}) - \boldsymbol{\nu}_1^T\boldsymbol{b}^{(1)} - \boldsymbol{\xi}^T\boldsymbol{x} - \epsilon\|\boldsymbol{\xi}\|_1 \tag{13}$$

such that

$$\boldsymbol{\nu}_K = -\boldsymbol{c},$$
$$\hat{\boldsymbol{\nu}}_k = \boldsymbol{W}^{(k+1)T}\boldsymbol{\nu}_{k+1} \ (k=K-1,\cdots,1),$$
$$\boldsymbol{\nu}_k = \boldsymbol{\alpha}_k \odot \hat{\boldsymbol{\nu}}_k \ (k=K-1,\cdots,1), \text{ and}$$
$$\boldsymbol{\xi} = \hat{\boldsymbol{\nu}}_1.$$

We decompose the first term in (13), and ignore the subscript $k$ as follows $\hat{\boldsymbol{\nu}}^T h(\hat{\boldsymbol{z}}) - (\boldsymbol{\alpha}\odot\hat{\boldsymbol{\nu}})^T\hat{\boldsymbol{z}}$. Further, we decompose this for each element, $\hat{\nu}h(\hat{z}) - \alpha\hat{\nu}\hat{z} = \hat{\nu}(h(\hat{z})-\alpha\hat{z})$. If the pre-activation bounds for $h$ are both positive (active ReLU), then $\alpha$ should be 1 not to make the inner supremum $> 0$. Similarly, if the pre-activation bounds for $h$ are both negative (dead ReLU), then $\alpha$ should be 0. In the case of unstable ReLU ($l \leq 0 \leq u$), if $\hat{\nu} < 0$, then we need to solve $\max_\alpha \inf_{\hat{z}} h(\hat{z}) - \alpha\hat{z}$. The inner infimum is 0 for $0 \leq \alpha \leq 1$, and is $< 0$ otherwise. On the other hand, if $\hat{\nu} \geq 0$, then we need to solve $\min_\alpha \sup_{\hat{z}} h(\hat{z}) - \alpha\hat{z}$. The inner supremum is $\max\{u-\alpha u, -\alpha l\}$, and thus the optimal dual variable is $\alpha^* = \frac{u}{u-l}$ which yields the optimal value (multiplied by $\hat{\nu}$) as $\hat{\nu}(u - \frac{u}{u-l}u) = -\frac{ul}{u-l}\hat{\nu}$ which is equivalent to using linear relaxation with $\overline{\boldsymbol{a}}\odot\boldsymbol{z}+\overline{\boldsymbol{b}} = \frac{u}{u-l}\odot(\boldsymbol{z}-\boldsymbol{l})$. We can represent it as $\overline{\boldsymbol{a}}\odot\boldsymbol{z}+\overline{\boldsymbol{b}} = \frac{\boldsymbol{u}^+}{\boldsymbol{u}^+-\boldsymbol{l}^-}\odot(\boldsymbol{z}-\boldsymbol{l}^-)$ to include the case of active/dead ReLU. For the lower linear bound $\underline{h}(\boldsymbol{z}) = \underline{\boldsymbol{a}}\odot\boldsymbol{z}+\underline{\boldsymbol{b}}$ in case of unstable ReLU, we can use any $\mathbf{0}\leq\underline{\boldsymbol{a}}\leq\mathbf{1}$ and $\underline{\boldsymbol{b}}=\mathbf{0}$ according to the dual relaxation with $\boldsymbol{\alpha}$. While CAP and CROWN-IBP use a dual feasible solution like $\boldsymbol{\alpha} = \frac{\boldsymbol{u}^+}{\boldsymbol{u}^+-\boldsymbol{l}^-}$ or $\boldsymbol{\alpha} = \mathbf{1}[\boldsymbol{u}^++\boldsymbol{l}^->0]$, our proposed method aims to optimize over the dual variable $\boldsymbol{\alpha}$ or equivalently optimize over $\mathbf{0}\leq\underline{\boldsymbol{a}}\leq\mathbf{1}$ to further tighten the upper bound on the loss.

# D Illustration of Linear Relaxations

Figure 4 provides some illustrations of linear relaxations used in IBP, CAP, CROWN-IBP, and the proposed method. CROWN-IBP adaptively chooses the relaxation variable so that the area between $\overline{h}$ and $\underline{h}$ is minimized. However, the smaller area does not necessarily imply the tighter bound, and the proposed method achieves tighter bounds than CROWN-IBP relaxation as shown in Figure 3.

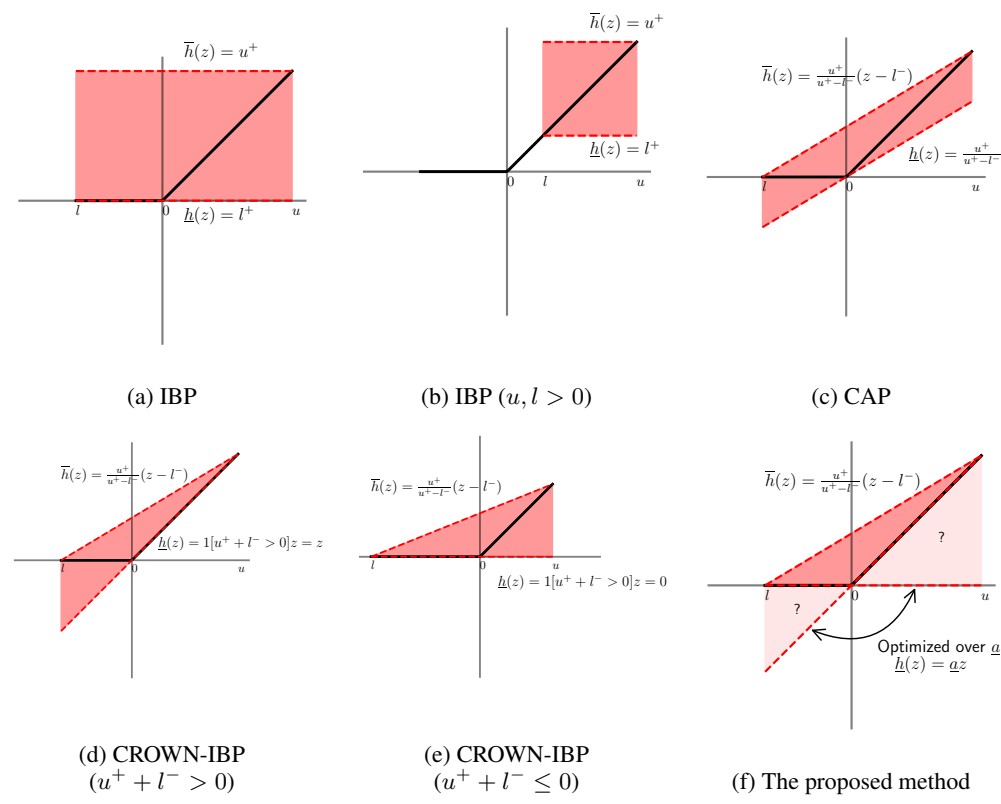

Figure 4: Illustrations of linear relaxation methods. Except for (b), they illustrate the relaxations when $l \leq 0 \leq u$ (Unstable ReLU). (b) Illustration of the relaxation of IBP when $u, l > 0$ (Active ReLU).

# E Learning curves for variants of CROWN-IBP

We find that $0.5/1$ and $1/1$ have less smooth loss landscapes than CROWN-IBP (as shown in Figure 5) where $p/q$ denotes the variant with sampling $\underline{a} \in \{0, 1\}$ with $P(\underline{a} = 1 \mid |l| > |u|) = p$ and $P(\underline{a} = 1 \mid |l| \leq |u|) = q$ for unstable ReLUs. On the other hand, $0/0$, $0/0.25$, and $0/0.5$ have more smooth loss landscape as in Figure 5, but they have looser bounds than CROWN-IBP.

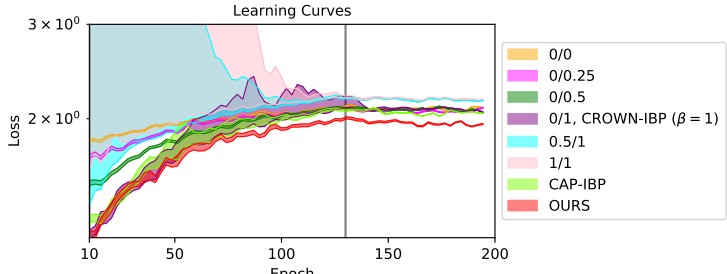

Figure 5: The learning curves for the scheduled value of $\epsilon$ with the loss variation along gradient descent direction (equivalent to Figure 1). As $\underline{a}$ becomes sparse, the loss variation is narrower.

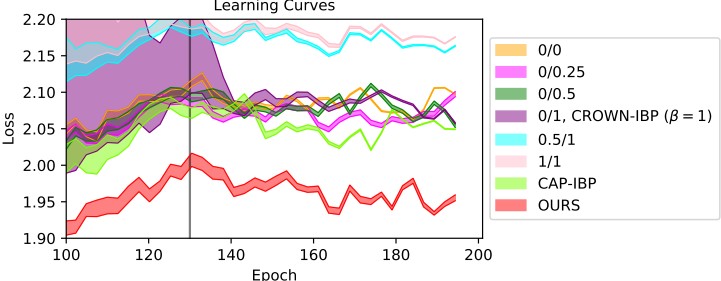

Figure 6: A zoomed-in version of Figure 5 for epochs 100-200.

Table 3: Performance (in terms of errors) of the variants of CROWN-IBP ($\beta = 1$). Note that $0/0.25$, $0/0.5$, and CAP-IBP start with looser bounds but they have more smooth landscape, which leads to a better performance than CROWN-IBP ($\beta = 1$) (highlighted with underline).

| Model | 0/0 | 0/0.25 | 0/0.5 | 0/1 CROWN-IBP ($\beta = 1$) | 0.5/1 | 1/1 | CAP-IBP | OURS |
|---|---|---|---|---|---|---|---|---|
| Standard | 70.66 | 64.50 | 62.72 | 63.24 | 70.69 | 71.41 | 60.36 | 57.14 |
| PGD | 73.84 | 72.67 | 71.42 | 71.70 | 76.68 | 77.03 | 69.46 | 66.88 |
| Verified | 77.60 | 74.47 | 74.92 | 75.72 | 78.38 | 78.73 | 74.29 | **71.45** |

# F  Proof

**Theorem 1.** *With gradient descent using a step size within an interval $I_t$ during the ramp-up period $(0 \leq \epsilon_t \leq \epsilon)$, the loss $\mathcal{L}^\epsilon$ for the target perturbation $\epsilon$ is reduced with*

$$\mathcal{L}^\epsilon(\boldsymbol{\theta}_{t+1}) \leq \mathcal{L}^\epsilon(\boldsymbol{\theta}_t)\big(1 - \frac{\mu}{2}\cos^2(\phi_t)\|\boldsymbol{H}_t^\epsilon \boldsymbol{u}_t\|^{-1}\big) \tag{4}$$

*for $\boldsymbol{u}_t = \frac{\nabla_{\boldsymbol{\theta}}\mathcal{L}^{\epsilon_t}(\boldsymbol{\theta}_t)}{\|\nabla_{\boldsymbol{\theta}}\mathcal{L}^{\epsilon_t}(\boldsymbol{\theta}_t)\|}$ where $0 < \mu \leq \frac{\|\nabla_{\boldsymbol{\theta}}\mathcal{L}^\epsilon\|^2}{2\mathcal{L}^\epsilon}$, $\cos(\phi_t) = \frac{\nabla_{\boldsymbol{\theta}}\mathcal{L}^{\epsilon T}\nabla_{\boldsymbol{\theta}}\mathcal{L}^{\epsilon_t}}{\|\nabla_{\boldsymbol{\theta}}\mathcal{L}^\epsilon\|\|\nabla_{\boldsymbol{\theta}}\mathcal{L}^{\epsilon_t}\|}$ and $\boldsymbol{H}_t^\epsilon$ satisfies $\mathcal{L}^\epsilon(\boldsymbol{\theta}_{t+1}) = \mathcal{L}^\epsilon(\boldsymbol{\theta}_t) + \nabla_{\boldsymbol{\theta}}\mathcal{L}^\epsilon(\boldsymbol{\theta}_t)^T\Delta_t + \frac{1}{2}\Delta_t^T\boldsymbol{H}_t^\epsilon\Delta_t$ and $\Delta_t^T\boldsymbol{H}_t^\epsilon\Delta_t > 0$ with $\Delta_t = \boldsymbol{\theta}_{t+1} - \boldsymbol{\theta}_t$.*

*Proof.*

$$\mathcal{L}^\epsilon(\boldsymbol{\theta}_{t+1}) = \mathcal{L}^\epsilon(\boldsymbol{\theta}_t) + \nabla_{\boldsymbol{\theta}}\mathcal{L}^\epsilon(\boldsymbol{\theta}_t)^T\Delta_t + \frac{1}{2}\Delta_t^T\boldsymbol{H}_t^\epsilon\Delta_t$$

$$= \mathcal{L}^\epsilon(\boldsymbol{\theta}_t) - \eta_t\nabla_{\boldsymbol{\theta}}\mathcal{L}^\epsilon(\boldsymbol{\theta}_t)^T\nabla_{\boldsymbol{\theta}}\mathcal{L}^{\epsilon_t}(\boldsymbol{\theta}_t) + \frac{1}{2}\eta_t^2\nabla_{\boldsymbol{\theta}}\mathcal{L}^{\epsilon_t}(\boldsymbol{\theta}_t)^T\boldsymbol{H}_t^\epsilon\nabla_{\boldsymbol{\theta}}\mathcal{L}^{\epsilon_t}(\boldsymbol{\theta}_t)$$

$$\leq \mathcal{L}^\epsilon(\boldsymbol{\theta}_t) - \frac{1}{\alpha}\frac{(\nabla_{\boldsymbol{\theta}}\mathcal{L}^\epsilon(\boldsymbol{\theta}_t)^T\nabla_{\boldsymbol{\theta}}\mathcal{L}^{\epsilon_t}(\boldsymbol{\theta}_t))^2}{2\nabla_{\boldsymbol{\theta}}\mathcal{L}^{\epsilon_t}(\boldsymbol{\theta}_t)^T\boldsymbol{H}_t^\epsilon\nabla_{\boldsymbol{\theta}}\mathcal{L}^{\epsilon_t}(\boldsymbol{\theta}_t)}$$

$$= \mathcal{L}^\epsilon(\boldsymbol{\theta}_t)\big(1 - \frac{1}{\alpha}\frac{(\nabla_{\boldsymbol{\theta}}\mathcal{L}^\epsilon(\boldsymbol{\theta}_t)^T\nabla_{\boldsymbol{\theta}}\mathcal{L}^{\epsilon_t}(\boldsymbol{\theta}_t))^2}{2\mathcal{L}^\epsilon(\boldsymbol{\theta}_t)}\frac{1}{\nabla_{\boldsymbol{\theta}}\mathcal{L}^{\epsilon_t}(\boldsymbol{\theta}_t)^T\boldsymbol{H}_t^\epsilon\nabla_{\boldsymbol{\theta}}\mathcal{L}^{\epsilon_t}(\boldsymbol{\theta}_t)}\big)$$

$$= \mathcal{L}^\epsilon(\boldsymbol{\theta}_t)\big(1 - \frac{1}{\alpha}\frac{(\nabla_{\boldsymbol{\theta}}\mathcal{L}^\epsilon(\boldsymbol{\theta}_t)^T\boldsymbol{u})^2}{2\mathcal{L}^\epsilon(\boldsymbol{\theta}_t)}\frac{1}{\boldsymbol{u}_t^T\boldsymbol{H}_t^\epsilon\boldsymbol{u}_t}\big)$$

$$\leq \mathcal{L}^\epsilon(\boldsymbol{\theta}_t)\big(1 - \frac{\mu}{\alpha}\cos^2(\phi_t)\frac{1}{\boldsymbol{u}_t^T\boldsymbol{H}_t^\epsilon\boldsymbol{u}_t}\big)$$

$$\leq \mathcal{L}^\epsilon(\boldsymbol{\theta}_t)\big(1 - \frac{\mu}{\alpha}\cos^2(\phi_t)\|\boldsymbol{H}_t^\epsilon\boldsymbol{u}_t\|^{-1}\big)$$

for any $\alpha > 1$ where in the first inequality a learning rate $\eta_t \in I_t \equiv [(1 - \sqrt{1 - \frac{1}{\alpha}})\eta_t^*, (1 + \sqrt{1 - \frac{1}{\alpha}})\eta_t^*]$ is used with $\eta_t^* = \frac{\nabla_{\boldsymbol{\theta}}\mathcal{L}^\epsilon(\boldsymbol{\theta}_t)^T\nabla_{\boldsymbol{\theta}}\mathcal{L}^{\epsilon_t}(\boldsymbol{\theta}_t)}{\nabla_{\boldsymbol{\theta}}\mathcal{L}^{\epsilon_t}(\boldsymbol{\theta}_t)^T\boldsymbol{H}_t^\epsilon\nabla_{\boldsymbol{\theta}}\mathcal{L}^{\epsilon_t}(\boldsymbol{\theta}_t)}$. And the last inequality comes from the Cauchy-Schwarz inequality. Using $\alpha = 2$, we can derive the final inequality (4).

$\square$

To prove Theorem 2, we first prove the following proposition. We note that $\boldsymbol{\theta}$ and $\boldsymbol{g}$ are vectorized and the matrix norm of Jacobian is naturally defined - for example, $\|\nabla_{\boldsymbol{\theta}}\boldsymbol{g}\|$ is induced by the vector norms defined in $\mathcal{X}$ and $\Theta$.

**Proposition 1.** *Given input $\boldsymbol{x} \in \mathcal{X}$ and perturbation radius $\epsilon$, let $M = \max\{\|\boldsymbol{x}'\| : \boldsymbol{x}' \in \mathbb{B}(\boldsymbol{x}, \epsilon)\}$. Then, for the upper bound $\overline{s}(\boldsymbol{x}; \boldsymbol{\theta}) = \max_{\boldsymbol{x}' \in \mathbb{B}(\boldsymbol{x}, \epsilon)} \boldsymbol{g}(\boldsymbol{x}; \boldsymbol{\theta})^T\boldsymbol{x}' + b(\boldsymbol{x}; \boldsymbol{\theta})$ with $b$ satisfying Assumption 1 (1), we have*

$$\|\nabla_{\boldsymbol{\theta}}\overline{s}(\boldsymbol{x}; \boldsymbol{\theta}_1) - \nabla_{\boldsymbol{\theta}}\overline{s}(\boldsymbol{x}; \boldsymbol{\theta}_2)\| \leq 2\epsilon\|\nabla_{\boldsymbol{\theta}}\boldsymbol{g}(\boldsymbol{x}; \boldsymbol{\theta}_{1,2})\| + M\|\nabla_{\boldsymbol{\theta}}\boldsymbol{g}(\boldsymbol{x}; \boldsymbol{\theta}_1) - \nabla_{\boldsymbol{\theta}}\boldsymbol{g}(\boldsymbol{x}; \boldsymbol{\theta}_2)\| + L_{\boldsymbol{\theta}\boldsymbol{\theta}}^b\|\boldsymbol{\theta}_1 - \boldsymbol{\theta}_2\| \tag{14}$$

*for any $\boldsymbol{\theta}_1, \boldsymbol{\theta}_2$, where $\boldsymbol{\theta}_{1,2}$ can be any of $\boldsymbol{\theta}_1$ and $\boldsymbol{\theta}_2$.*

*Proof.* Say $\overline{f}(\boldsymbol{x}'; \boldsymbol{\theta}) = \boldsymbol{g}(\boldsymbol{x}; \boldsymbol{\theta})^T\boldsymbol{x}' + b(\boldsymbol{x}; \boldsymbol{\theta})$ and the maximizer $\boldsymbol{x}_i^* = \arg\max_{\boldsymbol{x}' \in \mathbb{B}(\boldsymbol{x}, \epsilon)} \overline{f}(\boldsymbol{x}'; \boldsymbol{\theta}_i)$ for each $\boldsymbol{\theta}_i = \boldsymbol{\theta}_1, \boldsymbol{\theta}_2$. Then, we have

$$\|\nabla_{\boldsymbol{\theta}}\overline{s}(\boldsymbol{x}; \boldsymbol{\theta}_1) - \nabla_{\boldsymbol{\theta}}\overline{s}(\boldsymbol{x}; \boldsymbol{\theta}_2)\| = \|\nabla_{\boldsymbol{\theta}}\overline{f}(\boldsymbol{x}_1^*; \boldsymbol{\theta}_1) - \nabla_{\boldsymbol{\theta}}\overline{f}(\boldsymbol{x}_2^*; \boldsymbol{\theta}_2)\|$$

$$= \|\nabla_{\boldsymbol{\theta}}\overline{f}(\boldsymbol{x}_1^*; \boldsymbol{\theta}_1) - \nabla_{\boldsymbol{\theta}}\overline{f}(\boldsymbol{x}_2^*; \boldsymbol{\theta}_1) + \nabla_{\boldsymbol{\theta}}\overline{f}(\boldsymbol{x}_2^*; \boldsymbol{\theta}_1) - \nabla_{\boldsymbol{\theta}}\overline{f}(\boldsymbol{x}_2^*; \boldsymbol{\theta}_2)\|$$

$$\leq \|\nabla_{\boldsymbol{\theta}}\overline{f}(\boldsymbol{x}_1^*; \boldsymbol{\theta}_1) - \nabla_{\boldsymbol{\theta}}\overline{f}(\boldsymbol{x}_2^*; \boldsymbol{\theta}_1)\| + \|\nabla_{\boldsymbol{\theta}}\overline{f}(\boldsymbol{x}_2^*; \boldsymbol{\theta}_1) - \nabla_{\boldsymbol{\theta}}\overline{f}(\boldsymbol{x}_2^*; \boldsymbol{\theta}_2)\|. \tag{15}$$

The first term on the RHS can be upper bounded as follows:

$$\|\nabla_{\boldsymbol{\theta}}\overline{f}(\boldsymbol{x}_1^*; \boldsymbol{\theta}_1) - \nabla_{\boldsymbol{\theta}}\overline{f}(\boldsymbol{x}_2^*; \boldsymbol{\theta}_1)\| = \|\nabla_{\boldsymbol{\theta}}(\tilde{\boldsymbol{g}}_1^T\tilde{\boldsymbol{x}}_1^* - \tilde{\boldsymbol{g}}_1^T\tilde{\boldsymbol{x}}_2^*)\| = \|\nabla_{\boldsymbol{\theta}}(\boldsymbol{g}_1^T\boldsymbol{x}_1^* - \boldsymbol{g}_1^T\boldsymbol{x}_2^*)\|$$

$$= \|\nabla_{\boldsymbol{\theta}}\boldsymbol{g}_1(\boldsymbol{x}_1^* - \boldsymbol{x}_2^*)\|$$

$$\leq 2\epsilon\|\nabla_{\boldsymbol{\theta}}\boldsymbol{g}_1\|,$$

where $\boldsymbol{g}_i = \boldsymbol{g}(\boldsymbol{x};\boldsymbol{\theta}_i)$, $b_i = b(\boldsymbol{x};\boldsymbol{\theta}_i)$, $\tilde{\boldsymbol{g}}_i^T = [\boldsymbol{g}_i^T; b_i]$ and $\tilde{\boldsymbol{x}}^T = [\boldsymbol{x}^T; 1]$. And the second term on the RHS can be upper bounded as follows:

$$
\begin{aligned}
||\nabla_{\boldsymbol{\theta}}\overline{f}(\boldsymbol{x}_2^*;\boldsymbol{\theta}_1) - \nabla_{\boldsymbol{\theta}}\overline{f}(\boldsymbol{x}_2^*;\boldsymbol{\theta}_2)|| &= ||\nabla_{\boldsymbol{\theta}}(\tilde{\boldsymbol{g}}_1^T\tilde{\boldsymbol{x}}_2^* - \tilde{\boldsymbol{g}}_2^T\tilde{\boldsymbol{x}}_2^*)|| \\
&= ||\nabla_{\boldsymbol{\theta}}(\tilde{\boldsymbol{g}}_1 - \tilde{\boldsymbol{g}}_2)\tilde{\boldsymbol{x}}_2^*|| \\
&\leq ||\nabla_{\boldsymbol{\theta}}(\boldsymbol{g}_1 - \boldsymbol{g}_2)||||\boldsymbol{x}_2^*|| + ||\nabla_{\boldsymbol{\theta}}(b_1 - b_2)|| \\
&\leq M||\nabla_{\boldsymbol{\theta}}(\boldsymbol{g}_1 - \boldsymbol{g}_2)|| + L_{\boldsymbol{\theta}\boldsymbol{\theta}}^b||\boldsymbol{\theta}_1 - \boldsymbol{\theta}_2||,
\end{aligned}
$$

Therefore, we obtain

$$
\begin{aligned}
||\nabla_{\boldsymbol{\theta}}\overline{s}(\boldsymbol{x};\boldsymbol{\theta}_1) - \nabla_{\boldsymbol{\theta}}\overline{s}(\boldsymbol{x};\boldsymbol{\theta}_2)|| &\leq 2\epsilon||\nabla_{\boldsymbol{\theta}}\boldsymbol{g}_1|| + M||\nabla_{\boldsymbol{\theta}}(\boldsymbol{g}_1 - \boldsymbol{g}_2)|| + L_{\boldsymbol{\theta}\boldsymbol{\theta}}^b||\boldsymbol{\theta}_1 - \boldsymbol{\theta}_2|| \\
&= 2\epsilon||\nabla_{\boldsymbol{\theta}}\boldsymbol{g}(\boldsymbol{x};\boldsymbol{\theta}_1)|| + M||\nabla_{\boldsymbol{\theta}}\boldsymbol{g}(\boldsymbol{x};\boldsymbol{\theta}_1) - \nabla_{\boldsymbol{\theta}}\boldsymbol{g}(\boldsymbol{x};\boldsymbol{\theta}_2)|| + L_{\boldsymbol{\theta}\boldsymbol{\theta}}^b||\boldsymbol{\theta}_1 - \boldsymbol{\theta}_2||.
\end{aligned}
$$

Note that $\boldsymbol{\theta}_1$ in the first term is arbitrarily chosen in (15). Therefore, this leads to the final inequality (14). □

**Theorem 2.** *Suppose $\boldsymbol{x} \in \mathcal{X}$ is bounded $\|\boldsymbol{x}\| \leq M$ with some $M > 0$. For a linear relaxation-based method with the upper bound $\overline{s}_m(\boldsymbol{x};\boldsymbol{\theta}) = \max_{\boldsymbol{x}' \in \mathbb{B}(\boldsymbol{x},\epsilon)} \boldsymbol{g}_{\boldsymbol{\theta}}^{(m)}(\boldsymbol{x})^T\boldsymbol{x}' + b_{\boldsymbol{\theta}}^{(m)}(\boldsymbol{x})$, if each $b_{\boldsymbol{\theta}}^{(m)}$ and $\boldsymbol{p_s}$ satisfies Assumption 1, then*

$$
\begin{aligned}
&||\nabla_{\boldsymbol{\theta}}\mathcal{L}^\epsilon(\boldsymbol{\theta}_2) - \nabla_{\boldsymbol{\theta}}\mathcal{L}^\epsilon(\boldsymbol{\theta}_1)|| \\
&\leq \mathbb{E}_{(\boldsymbol{x},y)}\big[\max_m \big(2\epsilon||\nabla_{\boldsymbol{\theta}}\boldsymbol{g}_{\boldsymbol{\theta}_{1,2}}^{(m)}(\boldsymbol{x})|| + M||\nabla_{\boldsymbol{\theta}}\boldsymbol{g}_{\boldsymbol{\theta}_2}^{(m)}(\boldsymbol{x}) - \nabla_{\boldsymbol{\theta}}\boldsymbol{g}_{\boldsymbol{\theta}_1}^{(m)}(\boldsymbol{x})|| + L^{(m)}||\boldsymbol{\theta}_2 - \boldsymbol{\theta}_1||\big)\big] \quad (6)
\end{aligned}
$$

*for any $\boldsymbol{\theta}_1, \boldsymbol{\theta}_2$, where $L^{(m)} = L_{\boldsymbol{\theta}\boldsymbol{\theta}}^{b^{(m)}} + L_{\boldsymbol{\theta}}^{\boldsymbol{p_s}}||\nabla_{\boldsymbol{\theta}}\overline{s}(\boldsymbol{x};\boldsymbol{\theta}_{1,2})||$ and $\boldsymbol{\theta}_{1,2}$ can be any of $\boldsymbol{\theta}_1$ and $\boldsymbol{\theta}_2$.*

*Proof.* We start with the fact that the norm of the expected value of a random vector is smaller than expected norm of the random vector.

$$
\begin{aligned}
||\nabla_{\boldsymbol{\theta}}\mathcal{L}^\epsilon(\boldsymbol{\theta}_2) - \nabla_{\boldsymbol{\theta}}\mathcal{L}^\epsilon(\boldsymbol{\theta}_1)|| &= ||\nabla_{\boldsymbol{\theta}}\mathbb{E}_{(\boldsymbol{x},y)}[\mathcal{L}(\overline{s}(\boldsymbol{x};\boldsymbol{\theta}_2)] - \nabla_{\boldsymbol{\theta}}\mathbb{E}_{(\boldsymbol{x},y)}[\mathcal{L}(\overline{s}(\boldsymbol{x};\boldsymbol{\theta}_1)]|| \\
&\leq \mathbb{E}_{(\boldsymbol{x},y)}[||\nabla_{\boldsymbol{\theta}}\mathcal{L}(\overline{s}(\boldsymbol{x};\boldsymbol{\theta}_2) - \nabla_{\boldsymbol{\theta}}\mathcal{L}(\overline{s}(\boldsymbol{x};\boldsymbol{\theta}_1)||].
\end{aligned}
$$

We simplify the notation $\boldsymbol{p_s}$ as $\boldsymbol{p}$. Then we have

$$
\begin{aligned}
&||\nabla_{\boldsymbol{\theta}}\mathcal{L}(\overline{s}(\boldsymbol{x};\boldsymbol{\theta}_1)) - \nabla_{\boldsymbol{\theta}}\mathcal{L}(\overline{s}(\boldsymbol{x};\boldsymbol{\theta}_2))|| \\
&= ||\nabla_{\boldsymbol{\theta}}\overline{s}(\boldsymbol{x};\boldsymbol{\theta}_1)\nabla_{\overline{s}}\mathcal{L}(\overline{s}(\boldsymbol{x};\boldsymbol{\theta}_1)) - \nabla_{\boldsymbol{\theta}}\overline{s}(\boldsymbol{x};\boldsymbol{\theta}_2)\nabla_{\overline{s}}\mathcal{L}(\overline{s}(\boldsymbol{x};\boldsymbol{\theta}_2))|| \\
&= ||\sum_m \nabla_{\boldsymbol{\theta}}\overline{s}_m(\boldsymbol{x};\boldsymbol{\theta}_1)(\boldsymbol{p}_m(\boldsymbol{x};\boldsymbol{\theta}_1) - \delta_{y,m}) - \nabla_{\boldsymbol{\theta}}\overline{s}_m(\boldsymbol{x};\boldsymbol{\theta}_2)(\boldsymbol{p}_m(\boldsymbol{x};\boldsymbol{\theta}_2) - \delta_{y,m})|| \\
&= ||\nabla_{\boldsymbol{\theta}}\overline{s}(\boldsymbol{x};\boldsymbol{\theta}_1)(\boldsymbol{p}(\boldsymbol{x};\boldsymbol{\theta}_1) - \boldsymbol{e}^{(y)}) - \nabla_{\boldsymbol{\theta}}\overline{s}(\boldsymbol{x};\boldsymbol{\theta}_2)(\boldsymbol{p}(\boldsymbol{x};\boldsymbol{\theta}_2) - \boldsymbol{e}^{(y)})|| \\
&= ||\nabla_{\boldsymbol{\theta}}\overline{s}(\boldsymbol{x};\boldsymbol{\theta}_1)\boldsymbol{p}(\boldsymbol{x};\boldsymbol{\theta}_1) - \nabla_{\boldsymbol{\theta}}\overline{s}(\boldsymbol{x};\boldsymbol{\theta}_2)\boldsymbol{p}(\boldsymbol{x};\boldsymbol{\theta}_2)|| \\
&= ||\nabla_{\boldsymbol{\theta}}\overline{s}(\boldsymbol{x};\boldsymbol{\theta}_1)\boldsymbol{p}(\boldsymbol{x};\boldsymbol{\theta}_1) - \nabla_{\boldsymbol{\theta}}\overline{s}(\boldsymbol{x};\boldsymbol{\theta}_1)\boldsymbol{p}(\boldsymbol{x};\boldsymbol{\theta}_2) + \nabla_{\boldsymbol{\theta}}\overline{s}(\boldsymbol{x};\boldsymbol{\theta}_1)\boldsymbol{p}(\boldsymbol{x};\boldsymbol{\theta}_2) - \nabla_{\boldsymbol{\theta}}\overline{s}(\boldsymbol{x};\boldsymbol{\theta}_2)\boldsymbol{p}(\boldsymbol{x};\boldsymbol{\theta}_2)|| \\
&= ||\nabla_{\boldsymbol{\theta}}\overline{s}(\boldsymbol{x};\boldsymbol{\theta}_1)(\boldsymbol{p}(\boldsymbol{x};\boldsymbol{\theta}_1) - \boldsymbol{p}(\boldsymbol{x};\boldsymbol{\theta}_2)) + (\nabla_{\boldsymbol{\theta}}\overline{s}(\boldsymbol{x};\boldsymbol{\theta}_1) - \nabla_{\boldsymbol{\theta}}\overline{s}(\boldsymbol{x};\boldsymbol{\theta}_2))\boldsymbol{p}(\boldsymbol{x};\boldsymbol{\theta}_2)|| \\
&\leq ||\nabla_{\boldsymbol{\theta}}\overline{s}(\boldsymbol{x};\boldsymbol{\theta}_1)||||\boldsymbol{p}(\boldsymbol{x};\boldsymbol{\theta}_1) - \boldsymbol{p}(\boldsymbol{x};\boldsymbol{\theta}_2)|| + \max_m ||\nabla_{\boldsymbol{\theta}}\overline{s}_m(\boldsymbol{x};\boldsymbol{\theta}_1) - \nabla_{\boldsymbol{\theta}}\overline{s}_m(\boldsymbol{x};\boldsymbol{\theta}_2)|| \\
&\leq ||\nabla_{\boldsymbol{\theta}}\overline{s}(\boldsymbol{x};\boldsymbol{\theta}_1)||L_{\boldsymbol{\theta}}^{\boldsymbol{p}}||\boldsymbol{\theta}_1 - \boldsymbol{\theta}_2|| + \max_m ||\nabla_{\boldsymbol{\theta}}\overline{s}_m(\boldsymbol{x};\boldsymbol{\theta}_1) - \nabla_{\boldsymbol{\theta}}\overline{s}_m(\boldsymbol{x};\boldsymbol{\theta}_2)|| \\
&\leq \max_m \big(2\epsilon||\nabla_{\boldsymbol{\theta}}\boldsymbol{g}^{(m)}(\boldsymbol{x};\boldsymbol{\theta}_{1,2})|| + M||\nabla_{\boldsymbol{\theta}}\boldsymbol{g}^{(m)}(\boldsymbol{x};\boldsymbol{\theta}_1) - \nabla_{\boldsymbol{\theta}}\boldsymbol{g}^{(m)}(\boldsymbol{x};\boldsymbol{\theta}_2)|| + L^{(m)}||\boldsymbol{\theta}_1 - \boldsymbol{\theta}_2||\big).
\end{aligned}
$$

□

# G  Mode Connectivity

In this section, we check the mode connectivity [13] between two models that are trained using certifiable training methods. Mode connectivity is a framework that investigates the connectedness between two models by finding a high accuracy curve between those models. It enables us to understand the loss surface of neural networks.

Let $w_0$ and $w_1$ be two sets of weight corresponding to two different well-trained neural networks. Moreover, let $\phi_{\theta_c}(t)$ with $t \in [0, 1]$ be a continuous piece-wise smooth parametric curve with parameters $\theta_c$ such that $\phi_{\theta_c}(0) = w_0$ and $\phi_{\theta_c}(1) = w_1$. To find a low-loss path between $w_0$ and $w_1$, Garipov et al. [13] suggested to find the parameter $\theta_c$ that minimizes the expectation of a loss $\ell(w)$ over a distribution $q_{\theta_c}(t)$ on the curve,

$$L(\theta_c) = \mathbb{E}_{t \sim q_{\theta_c}(t)}[\ell(\phi_{\theta_c}(t)].$$

To optimize $L(\theta_c)$ for $\theta_c$, we use uniform distribution $U[0, 1]$ as $q_{\theta_c}(t)$ and Bezier curve [12] as $\phi_{\theta_c}(t)$, which provides a convenient parameterization of smoothness on the paths connecting two end points ($w_0$ and $w_1$) as follows:

$$\phi_{\theta_c}(t) = (1 - t)^2 w_0 + 2t(1 - t)\theta_c + t^2 w_1, 0 \leq t \leq 1.$$

A path $\phi_{\theta_c}$ is said to have a barrier if $\exists t$ such that $\ell(\phi_{\theta_c}(t)) > \max\{\ell(w_0), \ell(w_1)\}$. The existence of a barrier suggests the modes of two well-trained models are not connected by the path in terms of the given loss function $\ell$ [49].

We test the mode connectivity between the models trained with IBP, CROWN-IBP, and OURS. For example, to check the mode connectivity between two different models trained with CROWN-IBP and IBP, we use the loss function used on each model as a user-specified loss for training the parametric curve $\phi_{\theta_c}$. Therefore, we can obtain two curves as depicted in Figure 7, 8, and 9 for each pair of models. Here, we use the identical settings in Appendix A.1.

Figure 7 shows the mode-connectivity between CROWN-IBP and IBP. We use CROWN-IBP loss as user-specific loss in Figure 7a and IBP loss in Figure 7b. In this figure, we find that using CROWN-IBP loss (7a), there exists a barrier between the two models. This suggests they are not connected by the path in terms of CROWN-IBP loss. However, with IBP loss, there is no loss barrier separating the two models. This indicates that using CROWN-IBP, it is hard to optimize the parameters from $w_0$ to $w_1$, but IBP can.

Figure 8 shows the mode-connectivity results on IBP and OURS. We find that two models are not connected to each other using either IBP bound or OURS bound, since there exists a barrier in both curves. In this figure, we can also notify that OURS has tighter bounds than IBP because the value of the loss function using OURS is lower than that of IBP.

Finally, Figure 9 illustrates the mode connectivity between CROWN-IBP and OURS. Using CROWN-IBP as a user-specified loss function, we can find that the robust loss on the curve is higher than that of the end points. However, when OURS is used as a loss function, the robust loss generally decreases as the $t$ increases. It shows that OURS has much favorable loss landscape compared to CROWN-IBP. In addition, we can find that OURS has a tighter bound than CROWN-IBP, since the value of the robust loss using OURS is lower than CROWN-IBP.

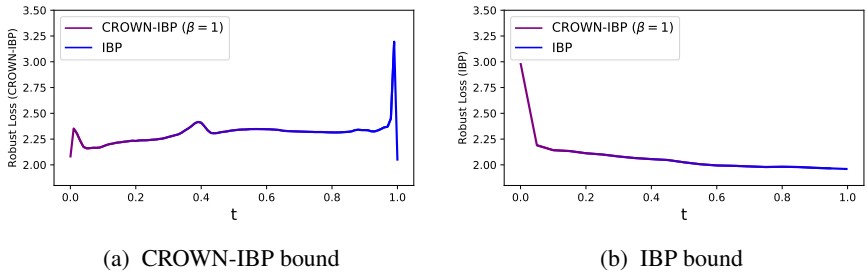

(a) CROWN-IBP bound

(b) IBP bound

Figure 7: Mode connectivity between CROWN-IBP and IBP, where $w_0$ and $w_1$ are well-trained models using CROWN-IBP bound and IBP bound, respectively. $\theta_c$ is trained using CROWN-IBP (7a) and IBP (7b), respectively.

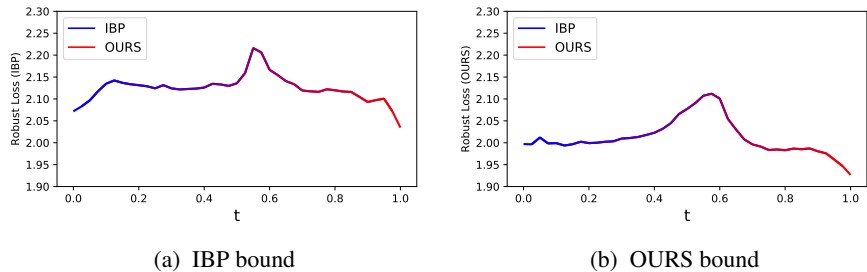

(a) IBP bound

(b) OURS bound

Figure 8: Mode connectivity between IBP and OURS, where $w_0$ and $w_1$ are well-trained models using IBP bound and OURS bound, respectively. $\theta_c$ is trained using IBP (8a) and OURS (8b), respectively.

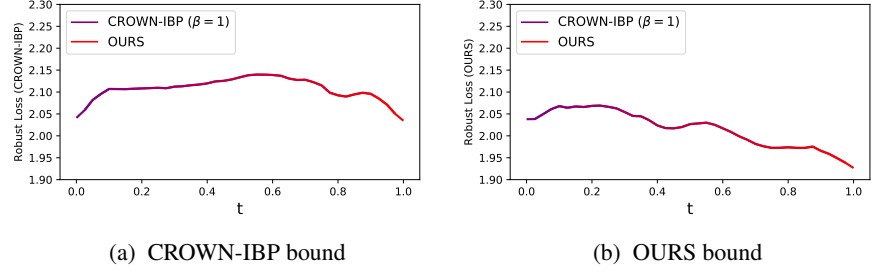

(a) CROWN-IBP bound

(b) OURS bound

Figure 9: Mode connectivity between CROWN-IBP and OURS, where $w_0$ and $w_1$ are well-trained models using CROWN-IBP bound and OURS bound, respectively. $\theta_c$ is trained using CROWN-IBP (9a) and OURS (9b), respectively.

# H ReLU

In this section, we investigate how pre-activation bounds $u$ and $l$ for the activation layer change during training. For each activation node, it is said to be "active" when the pre-activation bounds are both positive ($0 < l \leq u$), "unstable" when they span zero ($l \leq 0 \leq u$), and "dead" when they are both negative ($l \leq u < 0$).

Figure 10 shows the ratios of the number of active and dead ReLUs during the ramp-up period. Notably, we find that CROWN-IBP has more active ReLUs during training compared to the other three methods. Simultaneously, CROWN-IBP has the lowest ratio of dead ReLUs.

Figure 11 shows the numbers of active, unstable, and dead ReLUs during the ramp-up period. We find that in CROWN-IBP, the number of unstable and active ReLUs increases as the number of dead ReLUs decreases. This indicates that a number of dead ReLUs change to unstable ReLUs as the training $\epsilon$ increases. However, in the other methods, the number of unstable ReLUs is consistently small, while the number of active ReLUs decreases as the number of dead ReLUs increases.

Figure 12 depicts the histograms of the distribution of the slope $\frac{u^+}{u^+ - l^-}$ of the unstable ReLUs during the ramp-up period. In the early stages of CAP training, the slope distribution is concentrated around 0.4. However as the training progresses with a larger $\epsilon$, the histogram distribution moves to left, which indicates unstable ReLUs change to dead ReLUs. It is consistent with the results in Figure 11c. On the other hand, in the case of CROWN-IBP, the histogram distribution moves to right during training. It is the same with the results in Figure 11b, which shows that number of active ReLUs increases during training.

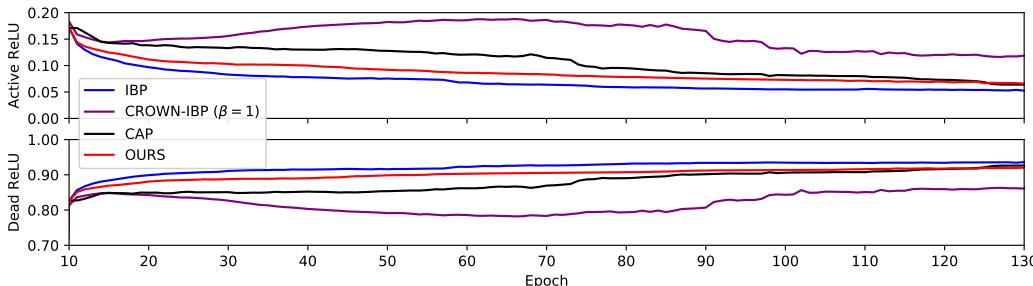

Figure 10: The ratio of the number of active (*top*) and dead (*bottom*) ReLUs during the ramp-up period.

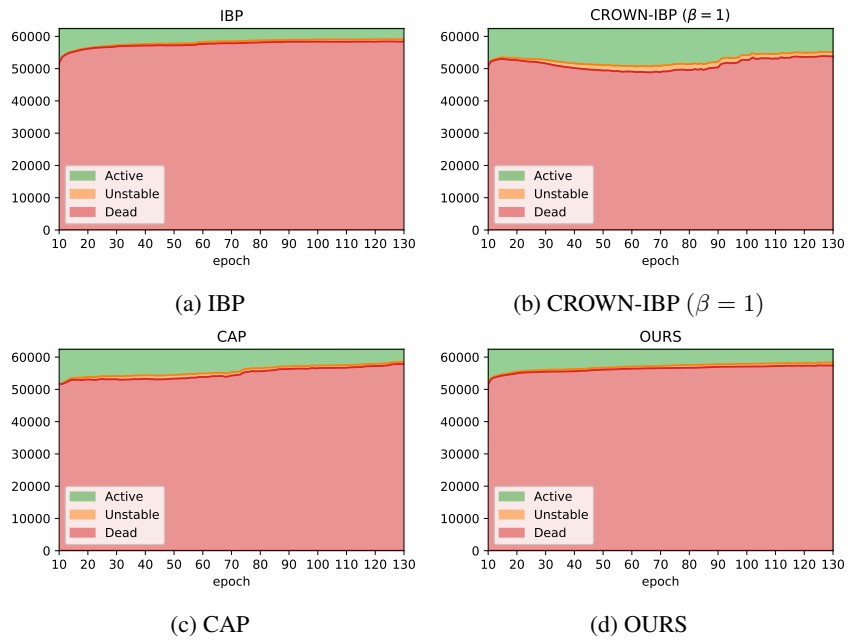

Figure 11: Number of active (Green), unstable (Orange), and dead (Red) ReLUs.

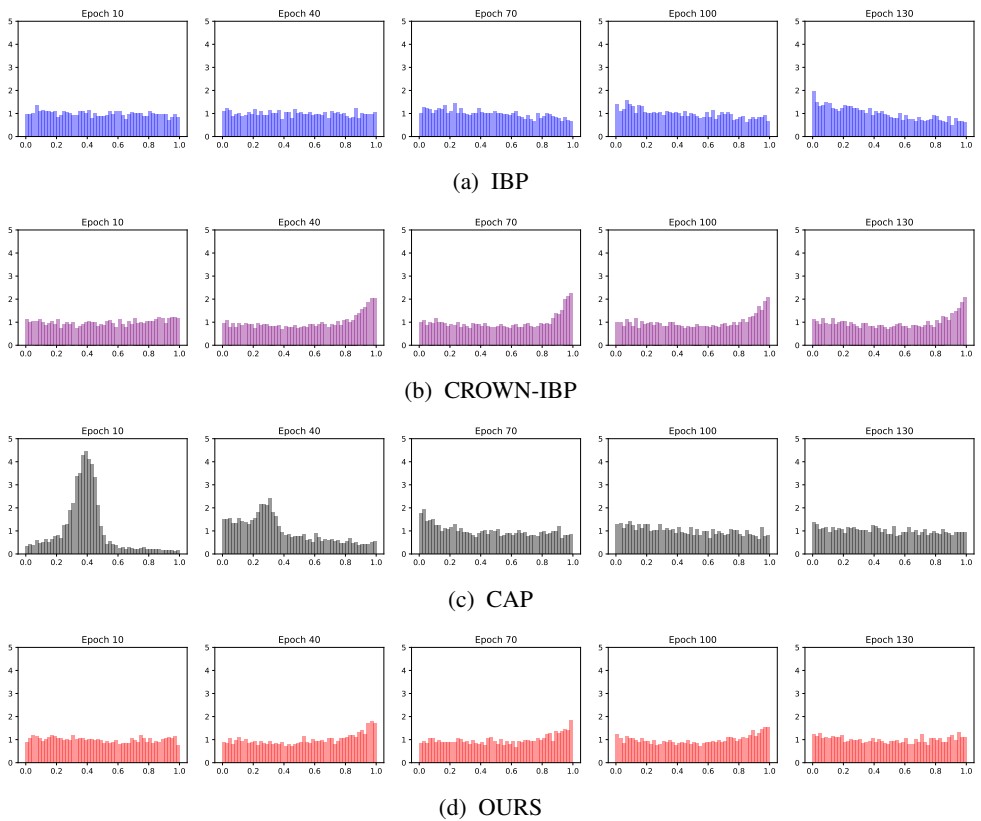

Figure 12: Histograms of the distribution of the slope $\frac{u^+}{u^+-l^-}$ when $l \leq 0 \leq u$ during the ramp-up period.

# I  $\beta$- and $\kappa$-schedulings

Table 4 shows the evaluation results of the models as in Table 1 but trained with different $\kappa$-scheduling (from 0 to 0). Table 5 shows the evaluation results of the proposed models trained with different $\kappa$- and $\beta$-schedulings.

Table 4: Test errors (Standard / PGD / Verified error) of IBP, CROWN-IBP ($\beta = 1$), CAP, and OURS on MNIST, CIFAR-10, and SVHN. See Appendix A for all the other settings, same as in Table 1. Bold and underline numbers are the first and second lowest verified error.

| Data | $\epsilon(l_\infty)$ | IBP | CROWN-IBP ($\beta = 1$) | CAP | OURS |
|------|------|------|------|------|------|
| MNIST | $\epsilon = 0.1$ | 1.25 / 2.31 / 3.10 | 1.23 / 2.19 / 2.75 | 0.80 / 1.73 / 3.19 | 1.09 / 1.86 / **2.28** |
| | $\epsilon = 0.2$ | 1.95 / 2.95 / 6.28 | 2.89 / 5.32 / 7.61 | 3.22 / 6.72 / 11.06 | 1.70 / 3.37 / **4.78** |
| | $\epsilon = 0.3$ | 3.67 / 5.55 / 9.74 | 6.11 / 11.33 / 17.51 | 19.19 / 35.84 / 47.85 | 3.39 / 4.85 / **9.12** |
| | $\epsilon = 0.4$ | 3.67 / 6.55 / 16.55 | 6.11 / 15.34 / 26.72 | - | 3.39 / 5.88 / **15.04** |
| CIFAR 10 | $\epsilon = 2/255$ | 43.60 / 52.62 / 56.58 | 32.15 / 42.67 / 49.36 | 28.80 / 38.95 / **48.50** | 32.04 / 43.13 / 49.62 |
| | $\epsilon = 4/255$ | 53.89 / 62.58 / 65.14 | 45.05 / 56.46 / 63.04 | 40.78 / 52.62 / 61.88 | 43.15 / 54.85 / **61.31** |
| | $\epsilon = 6/255$ | 61.37 / 68.64 / 70.82 | 53.87 / 65.03 / 71.08 | 49.20 / 60.85 / 69.03 | 50.99 / 62.23 / **67.59** |
| | $\epsilon = 8/255$ | 64.11 / 70.68 / 72.99 | 60.96 / 70.52 / 75.68 | 56.77 / 66.78 / 73.02 | 56.35 / 67.06 / **70.56** |
| | $\epsilon = 16/255$ | 69.74 / 76.66 / 79.86 | 79.14 / 83.64 / 84.36 | 75.11 / 80.67 / 82.07 | 66.96 / 75.63 / **78.08** |
| SVHN | $\epsilon = 0.01$ | 20.19 / 34.57 / 44.25 | 16.66 / 30.05 / 38.15 | 16.88/ 30.16 / **37.09** | 15.46 / 29.34 / 38.57 |

Table 5:  Test errors of OURS with different $\beta$- and $\kappa$-scheduling on MNIST and CIFAR-10.

| Data | $\epsilon(l_\infty)$ | OURS$_{1\to1}$ ($\kappa = 1 \to 0$) | | | OURS$_{1\to0}$ ($\kappa = 1 \to 0$) | | | OURS$_{1\to1}$ ($\kappa = 0 \to 0$) | | | OURS$_{1\to0}$ ($\kappa = 0 \to 0$) | | |
|------|------|------|------|------|------|------|------|------|------|------|------|------|------|
| | | Standard | PGD | Verfied | Standard | PGD | Verfied | Standard | PGD | Verfied | Standard | PGD | Verfied |
| MNIST | $\epsilon = 0.1$ | 1.09 | 1.77 | 2.36 | 1.29 | 2.29 | 3.58 | 1.09 | 1.86 | 2.28 | 1.15 | 2.03 | 3.53 |
| | $\epsilon = 0.2$ | 1.70 | 3.44 | 4.34 | 1.61 | 3.09 | 5.71 | 1.70 | 3.37 | 4.78 | 1.64 | 2.57 | 5.43 |
| | $\epsilon = 0.3$ | 3.49 | 5.59 | 9.79 | 2.42 | 4.37 | 7.84 | 3.39 | 4.85 | 9.12 | 2.44 | 4.41 | 8.00 |
| | $\epsilon = 0.4$ | 3.49 | 6.77 | 15.42 | 2.42 | 5.68 | 13.72 | 3.39 | 5.88 | 15.04 | 2.44 | 5.29 | 13.84 |
| CIFAR 10 | $\epsilon = 2/255$ | 31.49 | 42.73 | 49.42 | 37.77 | 48.30 | 54.43 | 32.04 | 43.13 | 49.62 | 38.58 | 48.59 | 54.63 |
| | $\epsilon = 8/255$ | 56.01 | 66.17 | 69.70 | 58.87 | 67.76 | 71.50 | 56.35 | 67.06 | 70.56 | 58.90 | 67.81 | 70.99 |
| | $\epsilon = 16/255$ | 65.39 | 75.39 | 77.87 | 66.24 | 74.69 | 78.66 | 66.96 | 75.63 | 78.08 | 66.76 | 75.17 | 77.99 |

## J    one-step vs multi-step

To get a tighter bound, we propose multi-step version of (7) as follows:

$$\underline{a}_{t+1} = \Pi_{[0,1]^n}\left(\underline{a}_t - \alpha\mathrm{sign}(\nabla_{\underline{a}}\mathcal{L}(\overline{s}(x, y, \epsilon; \theta, \phi), y))\right). \tag{16}$$

We compare the original 1-step method ($\alpha \geq 1$) to 7-step ($t = 7$) method with $\alpha = 0.1$. The results are summarized in Table 6. We found no significant difference between two methods even though multi-step takes multiple times with multi-step. Therefore, we decide to focus on one-step method.

Table 6: Test errors of OURS with different numbers of gradient update steps in (16) on CIFAR-10. Here, we use constant $\kappa = 0$.

| Data | $\epsilon(l_\infty)$ | OURS (1-step) | | | OURS (7-step) | | |
|---|---|---|---|---|---|---|---|
| | | Standard | PGD | Verfied | Standard | PGD | Verfied |
| **CIFAR-10** | $\epsilon = {}^2/_{255}$ | 32.04 | 43.11 | 49.62 | 31.40 | 42.30 | 49.20 |
| | $\epsilon = {}^8/_{255}$ | 56.35 | 67.03 | 70.56 | 54.44 | 66.29 | 71.53 |

## K    Train with $\epsilon_{train} \geq \epsilon_{test}$

### K.1    $\epsilon_{train} \geq \epsilon_{test}$ on MNIST

[46] and [15] observed that IBP performs better when using $\epsilon_{train} \geq \epsilon_{test}$ than $\epsilon_{train} = \epsilon_{test}$. Figure 7 shows the results with different $\epsilon_{train}$'s for each $\epsilon_{test}$. The overfitting issue is more prominent in the case of IBP and CROWN-IBP$_{1\rightarrow0}$ than the proposed method and CROWN-IBP$_{1\rightarrow1}$. However, using larger perturbations compromises the standard accuracy, and thus it is desirable to use smaller $\epsilon_{train}$.

Table 7: Comparison of the performance (Standard / PGD / Verified error) depending on various $\epsilon_{train}$. Here, we use constant $\kappa = 0$.

| Data | $\epsilon_{test}$ | $\epsilon_{train}$ | IBP | CROWN-IBP$_{1\rightarrow1}$ | OURS | CROWN-IBP$_{1\rightarrow0}$ |
|---|---|---|---|---|---|---|
| **MNIST** | 0.2 | 0.2 | 1.25 / 3.39 / 7.77 | 1.23 / 3.48 / 7.64 | 1.09 / 3.17 / 6.29 | 1.13 / 2.85 / 5.89 |
| | | 0.3 | 1.95 / 2.93 / 6.28 | 2.89 / 5.32 / 7.61 | 1.70 / 3.37 / 4.76 | 1.48 / 2.73 / 4.79 |
| | | 0.4 | 3.67 / 4.77 / 6.36 | 6.11 / 9.08 / 12.71 | 3.49 / 4.72 / 6.36 | 2.37 / 3.26 / 4.64 |
| | 0.3 | 0.3 | 1.95 / 3.31 / 12.90 | 2.89 / 7.35 / 14.97 | 1.70 / 4.82 / 9.20 | 1.48 / 3.52 / 9.40 |
| | | 0.4 | 3.67 / 5.55 / 9.74 | 6.11 / 11.33 / 17.51 | 3.49 / 5.59 / 9.79 | 2.37 / 3.63 / 7.22 |

### K.2    $\epsilon_{train} = 1.1\epsilon_{test}$ on CIFAR-10

As mentioned in Gowal et al. [15], we also train with $\epsilon_{train} = 1.1\epsilon_{test}$ on CIFAR-10. The results are shown in Table 8. They attain slightly improved performances in ${}^2/_{255}$, but not in ${}^8/_{255}$ and larger $\epsilon$.

Table 8: Comparison of the performance (Standard / PGD / Verified error) of the models trained with $\epsilon_{train}$ and $1.1\epsilon_{train}$. Here, we use constant $\kappa = 0$.

| Data | $\epsilon_{test}$ | $\epsilon_{train}$ | IBP | CROWN-IBP$_{1\rightarrow1}$ | OURS | CROWN-IBP$_{1\rightarrow0}$ |
|---|---|---|---|---|---|---|
| **CIFAR 10** | ${}^2/_{255}$ | ${}^2/_{255}$ | 43.6 / 52.71 / 56.58 | 32.15 / 42.67 / 49.36 | 32.04 / 43.13 / 49.62 | 37.25 / 47.19 / 52.53 |
| | | ${}^{2.2}/_{255}$ | 44.78 / 52.62 / 55.78 | 33.23 / 43.11 / 49.18 | 33.04 / 43.70 / 48.60 | 38.42 / 47.80 / 52.53 |
| | ${}^8/_{255}$ | ${}^8/_{255}$ | 64.11 / 70.68 / 72.99 | 60.96 / 70.52 / 75.68 | 56.35 / 67.06 / 70.56 | 56.95 / 67.89 / 70.43 |
| | | ${}^{8.8}/_{255}$ | 64.54 / 70.30 / 72.40 | 61.48 / 70.58 / 75.17 | 58.28 / 67.50 / 70.52 | 59.37 / 68.51 / 70.71 |

# L    Training time

All the training times are measured on a single TITAN X (Pascal) on Medium for CIFAR-10. We train with a batch size of 128 for OURS, CROWN-IBP$_{1\rightarrow1}$ and IBP, but with a batch size of 32 for CAP due to its high memory cost. For CAP, we use random projection of 50 dimensions.

- OURS: 115.9 sec / epoch
- CROWN-IBP$_{1\rightarrow1}$: 51.68 sec / epoch
- IBP: 14.85 sec / epoch
- CAP (batch size 32, 1 GPU): 751.0 sec / epoch
- CAP (batch size 64, 1 GPU): 724.6 sec / epoch
- CAP (batch size 128, 2 GPUs): 387.9 sec / epoch

# M   Loss and Tightness violin plots

We plot the equivalent tightness violin plots in Section 6 for models trained with other methods. The proposed method achieves the best results in terms of loss and tightness followed by CROWN-IBP, CAP-IBP, and RANDOM. Figure 13 (a)-(b), (c)-(d), and (e)-(f) show the tightness evaluated on the model trained by CROWN-IBP$_{1\to0}$, CROWN-IBP$_{1\to1}$ and IBP, respectively.

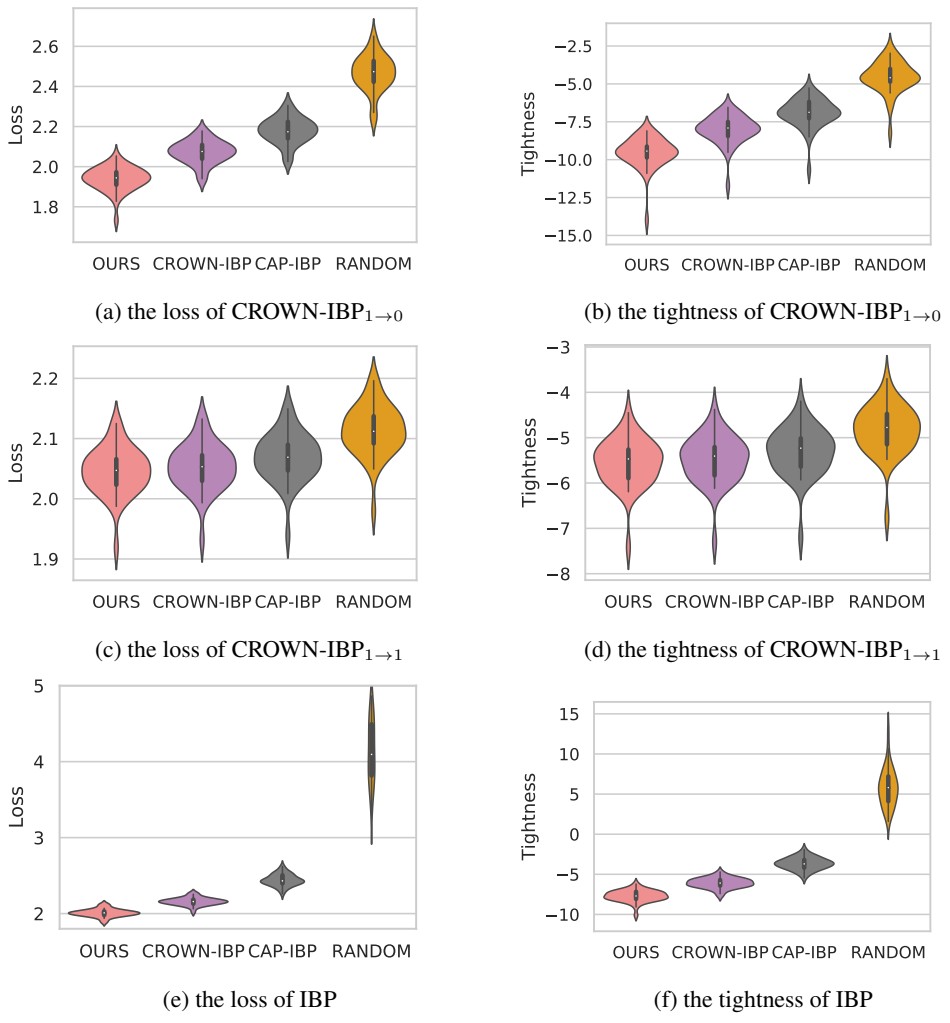

Figure 13: Violin plots of the test loss (*Left Column*) and of tightness (*Right Column*) for various linear relaxations same as in Section 6. Lower is better.

# N  Comparison with CAP-IBP

As in section E, we train a model with CAP-IBP and compare with the proposed method and CROWN-IBP ($\beta = 1$). Figure 14 shows that CAP-IBP has gradient differences larger than the proposed method and smaller than CROWN-IBP ($\beta = 1$), which leads to a performance between the proposed method and CROWN-IBP ($\beta = 1$) (see Table 3). CAP-IBP has looser bounds than CROWN-IBP ($\beta = 1$) as shown in Figure 3 and Figure 13, but with a relatively more smooth landscape, it can achieve a better performance than CROWN-IBP ($\beta = 1$).

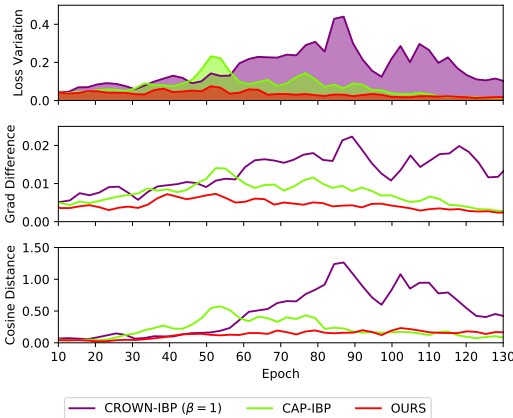

Figure 14: (Top) Loss variations along the gradient descent direction, (Middle) $\ell_2$-distance between two consecutive loss gradients and (Bottom) the cosine distance between them during the ramp-up phase.

# O    ReLU Stability

To see the effect of unstable ReLUs on smoothness, we adopt the ReLU stability loss (RS loss) $\mathcal{L}_{RS}(\boldsymbol{u}, \boldsymbol{l}) = -\tanh(1 + \boldsymbol{u} \cdot \boldsymbol{l})$ as a regularizer [42]. We use $\mathcal{L} + \lambda \mathcal{L}_{RS}$ as a loss and run CROWN-IBP ($\beta = 1$) with various $\lambda$ settings. We plot the smoothness and the tightness in Figure 15 and Figure 16 on $\lambda = 0$, $\lambda = 0.01$, $\lambda = 10$.

We found that small $\lambda$ suggested in Xiao et al. [42] has no effect on reducing the number of unstable ReLUs, and thus not on improving the smoothness as shown in Figure 11. By increasing $\lambda$, we observed that RS successfully reduces the number of unstable ReLUs with $\lambda = 10$. Figure 15 shows that large $\lambda$ leads to a smaller loss variation and gradient difference. This supports that unstable ReLUs are closely related to the smoothness of the loss landscape. However, as Xiao et al. [42] mentioned "placing too much weight on RS Loss can decrease the model capacity, potentially lowering the provable adversarial accuracy", the models trained with a large $\lambda \geq 1$ couldn't obtain a tightness of the upper bound and significant improvement on robustness as illustrated in Figure 16. The test errors (Standard / PGD / Verified) are 0.6278 / 0.7189 / 0.7634 on $\lambda = 0.01$ and 0.6090 / 0.7085 / 0.7600 on $\lambda = 10$.

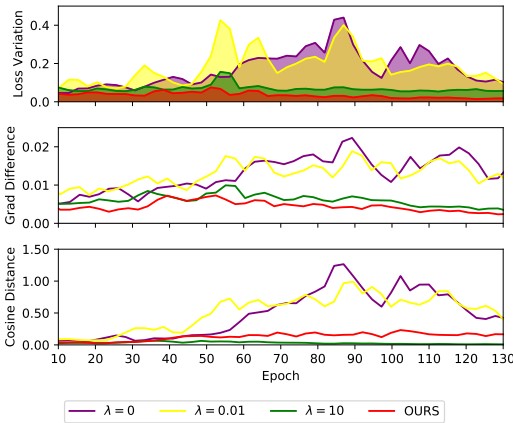

Figure 15: (Top) Loss variations along the gradient descent direction, (Middle) $\ell_2$-distance between two consecutive loss gradients and (Bottom) the cosine distance between them during the ramp-up phase on CROWN-IBP ($\beta = 1$) with $\lambda = 0$, $\lambda = 0.01$, $\lambda = 10$ and OURS.

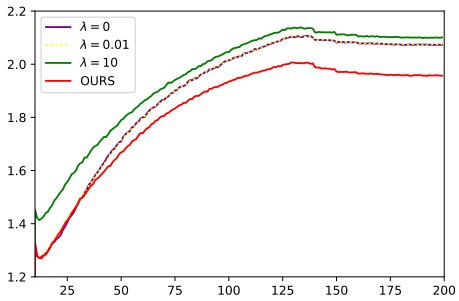

Figure 16: Robust loss of CROWN-IBP ($\beta = 1$) with $\lambda = 0$, $\lambda = 0.01$, $\lambda = 10$ and OURS during training.

# P    CBP$_{1 \to 0}$ and the smoothness

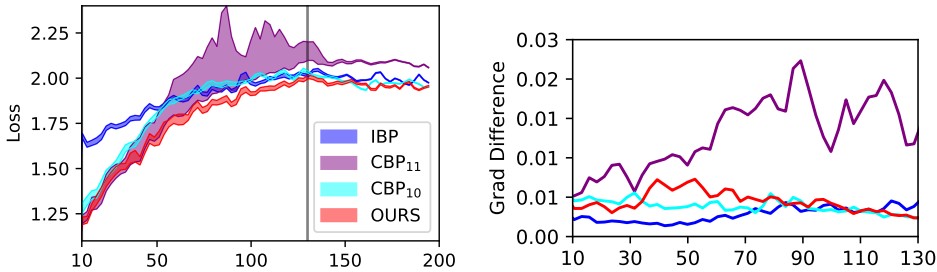

Figure 17: The success of CBP$_{1 \to 0}$ is also due to the smoothness. See the Figure 1 caption for more details.