# OpenReview forum: "Towards Better Understanding of Training Certifiably Robust Models against Adversarial Examples"
_NeurIPS.cc/2021/Conference — NeurIPS 2021 Poster_

### Official Review · Reviewer_Bv3w · 2021-07-11

**Rating:** 5
**Confidence:** 4

**Summary:**

This paper studies the problem of training certifiably robust models against adversarial examples. Besides the tightness of the upper bound, the paper identifies smoothness of the loss landscape as another key factor that influences the performance of certifiable training. Based on the theoretical analysis, the authors propose a new certifiable training method with the desired tightness and smoothness properties. Extensive experiments on MNIST, CIFAR-10, and SVHN show the effectiveness of the proposed method under certain range of perturbations.

**Limitations And Societal Impact:**

The authors have addressed the limitations in Section 7.

**Main Review:**

Originality: The paper did a good job in answering the question "What makes some loss landscapes more favorable than others?" While prior works mostly focus on developing tighter upper bound for the loss function, the paper shows that both tightness and smoothness of the upper bound are important for linear relaxation based methods.

Quality: The main results of Theorems 1 and 2 are interesting. I am glad to see the analysis also inspires an algorithm which improves SOTA by experiments. One of the major drawbacks of this paper is that, the authors fail to connect Theorems 1 and 2 to the certified robustness accuracy.

Clarity: The paper is well-written and easy to understand its main message.

Significance: The significance of this paper is on average. Though the paper achieves SOTA results in certain cases, the improvement is marginal (roughly 1%-2% improvement). So it seems that the extra smoothness consideration is not so important.

**Time Spent Reviewing:**

4

---

> ### Author Response · Authors · 2021-08-10
> **Authors' response**
>
> We thank the reviewers for their valuable comments (C).  Please find our responses (R) below:
>
> - C1: One of the major drawbacks of this paper is that, the authors fail to connect Theorems 1 and 2 to the certified robustness accuracy.
>     - R1: We may use the concept of a calibration function to link the loss function and the accuracy (risk) [1], which is out of scope of the paper.
>     - However, we think it is more important to consider the loss because our focus is to understand the effect of the smoothness on the optimization behaviors.
>     - [1] How to compare different loss functions and their risks.
>
> - C2: Significance: The significance of this paper is on average. Though the paper achieves SOTA results in certain cases, the improvement is marginal (roughly 1%-2% improvement). So it seems that the extra smoothness consideration is not so important.
>     - R2: To summarize, IBP and CBP$\_{1\rightarrow0}$ are much worse ($\Delta$=-7.59,-4.37) than the proposed method in a small epsilon regime ($\epsilon=2/255$) because they lack the tightness, and CBP$\_{1\rightarrow1}$ is much worse ($\Delta$=-4.90) than the proposed method in a large epsilon regime ($\epsilon=8/255$) because CBP$\_{1\rightarrow1}$ lacks the smoothness. The proposed method has the two desired properties: (1) the smoothness of IBP and CBP$_{1\rightarrow0}$ and (2) the tightness of CROWN (CBP$\_{1\rightarrow1}$), without introducing the mixture parameter $\beta$ and it shows a decent performance in a wide range of adversarial perturbations.
>     - Moreover, we again emphasize that our goal is rather to understand the effect of the smoothness and the tightness in a certain regime of the epsilon than to beat the SOTA methods.

---

### Official Review · Reviewer_fj3F · 2021-07-16

**Rating:** 3
**Confidence:** 5

**Summary:**

It has been empirically observed that using tighter relaxations of loss function (largely based on the linear relaxation techniques [1,
2, 3]) do not result in improved performance for certifiably robust training compared to Interval bound propagation techniques which use a much looser relaxation [4, 5].


[1] E. Wong and Z. Kolter. Provable defenses against adversarial examples via the convex outer adversarial polytope.
[2] E. Wong, F. Schmidt, J. H. Metzen, and J. Z. Kolter. Scaling provable adversarial defenses.
[3] K. Dvijotham, S. Gowal, R. Stanforth, R. Arandjelovic, B. O’Donoghue, J. Uesato, and P. Kohli. Training verified learners with learned verifiers.
[4] M. Mirman, T. Gehr, and M. Vechev. Differentiable abstract interpretation for provably robust neural networks.
[5] S. Gowal, K. Dvijotham, R. Stanforth, R. Bunel, C. Qin, J. Uesato, T. Mann, and P. Kohli. On the effectiveness of interval bound propagation for training verifiably robust models.


**Limitations And Societal Impact:**

I see no negative societal impact of this work.

**Main Review:**

The paper studies this problem through the lens of loss landscape.

I believe the paper is not well written. There are some things which are not clearly defined or explained. It is not clear to me what the authors exactly mean by the loss landscape. It can be defined in many different ways: largest eigenvalues of the Hessian, largest principal curvatures of the manifold with loss = constant, or increase in the function value in the direction of the gradient. I have read Figure 1 where they use 3 different metrics as a heuristic for the loss landscape but in many different locations in the paper where the authors refer to loss landscape, what exactly do they mean by it? Which of these metrics are they using to describe loss landscape?  Moreover, it is never formally discussed anywhere whether the loss landscape is calculated with respect to the input or the weights of the neural network. Ideally, for robustness analysis, the loss landscape should be discussed with respect to the inputs (assuming weights of the neural network are constant).

The results are not very strong. For some epsilon values, their method works better but for others the previous methods significantly outperform. Moreover, in order to establish the generality of results comparison should also be performed for the l_{2} threat model.

**Time Spent Reviewing:**

6

---

> ### Author Response · Authors · 2021-08-10
> **Authors' response**
>
> We thank the reviewers for their valuable comments (C).  Please find our responses (R) below:
> - C1: I believe the paper is not well written. There are some things which are not clearly defined or explained. It is not clear to me what the authors exactly mean by the loss landscape. It can be defined in many different ways: largest eigenvalues of the Hessian, largest principal curvatures of the manifold with loss = constant, or increase in the function value in the direction of the gradient.
> - I have read Figure 1 where they use 3 different metrics as a heuristic for the loss landscape but in many different locations in the paper where the authors refer to loss landscape, what exactly do they mean by it? Which of these metrics are they using to describe loss landscape?
>     - R1: We think the loss landscape is widely-used term (cf. [1]). We used it for the surface (graph) of the loss function with respect to the parameters, $\mathcal{G}$={$(\theta, \mathcal{L}(\theta)): \theta\in\mathbb{R}^m$} or for the loss function $\mathcal{L}(\theta)$ itself. We will clarify this in the revised manuscript.
>    - It is true that the smoothness of the loss landscape can be defined in many different ways: we measured the smoothness with three different metric: Loss Variation, Grad Difference, and Cosine Distance as detailed in Appendix A.1. We will move these definition to the main text.
>     - Loss Variation: $\mathcal{L}^{\epsilon_t}(\theta([0,5]))$ where $\theta(\lambda)=\theta_t-\lambda\eta\nabla_\theta\mathcal{L}^{\epsilon_t}(\theta_t)$
>     - Grad Difference: $||\mathcal{L}^{\epsilon_t}(\theta_t)-\mathcal{L}^{\epsilon_t}(\theta_{t+1})||$
>     - Cosine Difference: $1-\cos(\mathcal{L}^{\epsilon_t}(\theta_t),\mathcal{L}^{\epsilon_t}(\theta_{t+1}))$ where $\cos(v_1,v_2)$ is the cosine value of the angle between two vectors $v_1$ and $v_2$.
>     - They are highly related with each other. Loss Variation is similar to "the increase in the function value in the direction of the gradient" and Grad Difference is approximated with the effective curvature of the Hessian in the direction of $\Delta_t$ as shown in Eq. (5) which is similar to "the largest eigenvalues of the Hessian". In the theoretical analysis, we usually used Grad Difference as in (5) and (6) in theorem 2.
>     - [1] Visualizing the loss landscape of neural nets
>
>
> - C2: Moreover, it is never formally discussed anywhere whether the loss landscape is calculated with respect to the input or the weights of the neural network. Ideally, for robustness analysis, the loss landscape should be discussed with respect to the inputs (assuming weights of the neural network are constant).
>     - R2: (cf. R1) The loss landscape is the (training) loss function with respect to the parameters.
>     - In line 107, we fix the training set and denote the empirical loss $\mathbb{E}_{(x,y)}[\mathcal{L}(\bar{s}(x,y,\epsilon;\theta), y)]$ as $\mathcal{L}^\epsilon(\theta)$, i.e, a function of the parameters $\theta$.
>     - We analyze the smoothness as a factor that influences the certifiable training in **the optimization sense**. It is natural to deal the loss function with respect to the weights.
>     - In details, theorem 1 says that "the smooth loss function is more favorable for the optimization".
>
> - C3: The results are not very strong. For some epsilon values, their method works better but for others the previous methods significantly outperform.
>     - R3: To summarize, IBP and CBP$\_{1\rightarrow0}$ are much worse ($\Delta$=-7.59,-4.37) than the proposed method in a small epsilon regime ($\epsilon=2/255$) because they lack the tightness, and CBP$\_{1\rightarrow1}$ is much worse ($\Delta$=-4.90) than the proposed method in a large epsilon regime ($\epsilon=8/255$) because CBP$\_{1\rightarrow1}$ lacks the smoothness. The proposed method has the two desired properties: (1) the smoothness of IBP and CBP$_{1\rightarrow0}$ and (2) the tightness of CROWN (CBP$\_{1\rightarrow1}$), without introducing the mixture parameter $\beta$ and it shows a decent performance in a wide range of adversarial perturbations.
>     - Moreover, we again emphasize that our goal is rather to understand the effect of the smoothness and the tightness in a certain regime of the epsilon than to beat the SOTA methods.

---

### Official Review · Reviewer_iUZ4 · 2021-07-16

**Rating:** 4
**Confidence:** 5

**Summary:**

This paper investigates certified training with different linear relaxations.
The authors provide several theoretical and experimental results which can suggest that
some relaxations produce non-smooth loss landscape, and result in bad performance.
Based on this, they also propose a new certified defense and evaluate it on several benchmarks.


**Limitations And Societal Impact:**

I could not find discussion of societal impact.

**Main Review:**

In general, I think that investigating underlying reasons why some linear relaxations perform worse than others is important research direction. However, I feel this paper falls short of delivering satisfying explanation, while at the same time newly proposed certified defense achieves worse results than prior work. I list my three main concerns below.

If we view this paper as a new certified defense, then the problem is that results are still quite a bit below the SOTA numbers from prior work. For example, on CIFAR-10 with 8/255 CROWN-IBP achieves ~3% higher verified accuracy, and with 2/255 COLT achieves ~8% higher. Given that this paper focuses quite a bit on improving certified defense, I think the final results are disappointing as we see no improvement over prior work.

I have some concerns with the results presented in Section 4.2. The main idea of Theorem 1 is to upper bound loss in the next step using difference in gradients, and then in Theorem 2 the gradient difference is bounded by assuming Lipschitz continuity.
I am not sure whether considering upper bound is very informative here (e.g. all losses could be 0 and inequalities would hold).
Instead, I think more interesting result would be to lower bound the loss of non-smooth relaxations, which would actually
prove that they are not going to work well for training. Also, Equation 5 approximates Hessian vector product by the difference in gradients, and it is not clear how good is this approximation so we do not even get formal upper bound right now.

There is also a concurrent work of Jovanovic et al. [1] which proposes different hypothesis that bad performance of linear
relaxations is due to the discontinuity and sensitivity of these relaxations. Could the authors discuss what is the relationship
between the results presented in this submission and the work of Jovanovic et al.?
In particular, I am curious about Assumption 1 where authors assume that linear relaxation produces Lipschitz continuous loss (which implies continuity) while on the contrary Jovanovic et al. show that some relaxations are discontinuous (e.g CROWN-IBP) and they claim that this is the cause of bad performance. Because of these assumptions, it seems that theorems in this submission cannot be applied to discontinuous relaxations, so they cannot explain their bad performance. Could authors discuss the relationship between these two works?

L120: seems there is a symbol missing after g^T *

L164: increases -> increase

[1] Jovanović, Nikola, et al. "Certified Defenses: Why Tighter Relaxations May Hurt Training?." arXiv preprint arXiv:2102.06700 (2021).


**Time Spent Reviewing:**

5

---

> ### Author Response · Authors · 2021-08-10
> **Authors' response**
>
> In general, I think that investigating underlying reasons why some linear relaxations perform worse than others is important research direction. However, I feel this paper falls short of delivering satisfying explanation, while at the same time newly proposed certified defense achieves worse results than prior work. I list my three main concerns below.
>
> - C1: If we view this paper as a new certified defense, then the problem is that results are still quite a bit below the SOTA numbers from prior work. For example, on CIFAR-10 with 8/255 CROWN-IBP achieves ~3% higher verified accuracy, and with 2/255 COLT achieves ~8% higher. Given that this paper focuses quite a bit on improving certified defense, I think the final results are disappointing as we see no improvement over prior work.
>     - R1: To summarize, IBP and CBP$\_{1\rightarrow0}$ are much worse ($\Delta$=-7.59,-4.37) than the proposed method in a small epsilon regime ($\epsilon=2/255$) because they lack the tightness, and CBP$\_{1\rightarrow1}$ is much worse ($\Delta$=-4.90) than the proposed method in a large epsilon regime ($\epsilon=8/255$) because CBP$\_{1\rightarrow1}$ lacks the smoothness. The proposed method has the two desired properties: (1) the smoothness of IBP and CBP$_{1\rightarrow0}$ and (2) the tightness of CROWN (CBP$\_{1\rightarrow1}$), without introducing the mixture parameter $\beta$ and it shows a decent performance in a wide range of adversarial perturbations.
>     - It is not true that CROWN-IBP achieves ~3% higher verified accuracy. CROWN-IBP (CBP$_{10}$) has 1.5%p higher verified accuracy but 3.24/2.84%p lower standard/pgd accuracy than the proposed method.
>     - As COLT and CAP use much tighter bound, it is natural for them to have better verfiied accuracy when $\epsilon$ is small, but the smoothness plays more important role than the tightness in a large epsilon regime.
>     - However, we again emphasize that our goal is rather to understand the effect of the smoothness and the tightness in a certain regime of the epsilon than to beat the SOTA methods.
>
> - C2: I have some concerns with the results presented in Section 4.2. The main idea of Theorem 1 is to upper bound loss in the next step using difference in gradients, and then in Theorem 2 the gradient difference is bounded by assuming Lipschitz continuity. I am not sure whether considering upper bound is very informative here (e.g. all losses could be 0 and inequalities would hold). Instead, I think more interesting result would be to lower bound the loss of non-smooth relaxations, which would actually prove that they are not going to work well for training. Also, Equation 5 approximates Hessian vector product by the difference in gradients, and it is not clear how good is this approximation so we do not even get formal upper bound right now.
>     - R2: The suggested lower bound would prove that the method with non-smooth landscapes are not going to work well for training, but this does not prove whether smooth methods are going to work well or not. Instead, we prove that the method with smooth landscapes are going to work well in the optimization, which can provide an answer to the question: "Why IBP outperforms linear relaxation-based methods?" and "What factors help the performance of certifiable training?".
>
> - C3: There is also a concurrent work of Jovanovic et al. [1] which proposes different hypothesis that bad performance of linear relaxations is due to the discontinuity and sensitivity of these relaxations. Could the authors discuss what is the relationship between the results presented in this submission and the work of Jovanovic et al.? In particular, I am curious about Assumption 1 where authors assume that linear relaxation produces Lipschitz continuous loss (which implies continuity) while on the contrary Jovanovic et al. show that some relaxations are discontinuous (e.g CROWN-IBP) and they claim that this is the cause of bad performance. Because of these assumptions, it seems that theorems in this submission cannot be applied to discontinuous relaxations, so they cannot explain their bad performance. Could authors discuss the relationship between these two works?
>     - R3: Thank you for pointing out the reference. The concepts of discontinuity and sensitivity together seem highly related to the smoothness, and the arguments of the two studies seem to support each other.
>     - We measure the smoothness with several metrics, e.g. $||\nabla\_\theta \mathcal{L}^{\epsilon\_t}(\theta_{t+1}) - \nabla\_\theta \mathcal{L}^{\epsilon\_t}(\theta_{t})||$ as in Eq. (5).
>     - We also think the discontinuity can be the cause of bad performance since it natually has a bad smoothness.
>     - We understand that many linear relaxation methods are discontinuous with respect to the weight, and thus the Lipschitzness assumptions are no longer applied.
>     - **Still, theorem 1 can explain their bad performance** with the observation in Figure 1 that they have non-smooth landscapes with large $||\nabla\_\theta \mathcal{L}^{\epsilon\_t}(\theta_{t+1}) - \nabla\_\theta \mathcal{L}^{\epsilon\_t}(\theta_{t})||$ .
>     - While theorem 2 is proved under the assumption 1, it can be easily modified **without the assumption but with the messy terms**, i.e., $||\nabla\_\theta b(x;\theta\_2) - \nabla\_\theta b(x;\theta\_1)||+||p(x;\theta\_2) - p(x;\theta\_1)||||\nabla_\theta \bar{s}(x;\theta_{1,2})||$ instead of $L^{(m)}||\theta\_2-\theta\_1||$.
>     - The modified inequality would prove that the discontinuous methods have large $||\nabla\_\theta \mathcal{L}^{\epsilon}(\theta_{2}) - \nabla\_\theta \mathcal{L}^{\epsilon}(\theta_{1})||$ since they also have large $||\nabla\_\theta b(x;\theta\_2) - \nabla\_\theta b(x;\theta\_1)||$ and $||p(x;\theta\_2) - p(x;\theta\_1)||$ compared to the continuous ones.
>
> - C4: L120: seems there is a symbol missing after g^T *; L164: increases -> increase
>     - Thank you for pointing out the typos. We will change "increases" to "increase" in L164.
>     - We write "... $h_i^{[k]}(\cdot)$ can be uppper bounded by a linear function $g^T\cdot+b$ over $\mathbb{B}(x,\epsilon)$".
>     - This is because the variable $x$ is used with different meaning in the same sentence.
>     - We will revise it as "...$h_i^{[k]}(\cdot)$ can be uppper bounded by a linear function $h_i^{[k]}(\xi)\leq g^T\xi+b$ over $\mathbb{B}(x,\epsilon)$".

---

### Official Review · Reviewer_fogS · 2021-07-16

**Rating:** 4
**Confidence:** 4

**Summary:**

The authors study the loss landscape of certifiably robust training methods such as IBP or CROWN. They show that the smoothness of the loss landscape impact the robustness obtained, independent of the tightness of the used bound. They propose a method based on CROWN-IBP to improve smoothness and show that this also leads to slightly better certified robustness.

**Limitations And Societal Impact:**

See above.

**Main Review:**

Strengths:
- Good related work section outlining early and previous work.
- Separate section to introduce notation (which is complex in this paper).
- Both empirical and theoretical analysis of the loss landscape.
- Visually nice and easy-to-read plots.
- Simple and well-motivated method to improve smoothness.
- Comparison to multiple methods on several datasets in the experiments.
- Empirical validation that the proposed method also improves smoothness.

Weaknesses:
- The authors start by referring to “more favorable loss landscape”. However, from the paper it mostly refers to being more smooth. This happens a couple of times throughout the paper and can be confusing – I would prefer being precise and using smoothness throughout the paper.
- Generally, the paper is incredibly dense in terms of notation and math. Most equations/theorems/definitions are not explained “in words” and following the paper requires jumping back and forth a lot. Many aspects of notation are unclear (see below) and I am pretty sure that the paper is hard to understand without being familiar with previous work (i.e., IBP, CROWN, etc.). Even if this is the case, one has to go back to these papers to follow the submitted one.
- Regarding the empirical smoothness measures: the authors refer to [29] for computing loss variation. This should be explained in the paper as it seems to be an important point of the paper.
- I am also missing some discussion of why these measures correlate with loss landscape smoothness, i.e., what the intuition is.

- Comments on structure and high-level content:
– The examples in line 235 should come earlier. I essentially needed to read 5 pages before making the exact connection between the bounds a and the approximate gradient g.
– It would make sense to discuss theorem 1 first, this clearly motivates why the authors empirically look at the difference of gradients.
– After section 3, I had to read through a lot of notation, without really knowing why. The authors should try to give me a high-level impression (i.e., that they phrase CROWN-IBP in their notation and later try to improve upon it by changing the bounds).

- Comments on notation:
– Line 100: e^{(y)} is not clearly introduced, meaning the reader has to stop and try to understand how exactly C(y) is defined.
– Line 120: g^T \cdot is slightly unclear what the \cdot refers to. Generally, in the discussion around Definition 1 it is unclear how exactly g is based on the function bounds. This becomes slightly clearer when discussing IBP, but not to an extent that the paper can be read on its own (without checking related papers).
– Line 133ff: the introduction of IPB and CROWN-IBP requires the reader to be very familiar with these methods, otherwise it seems extremely hard to follow the discussion. I would prefer to introduce one of these methods (e.g, CROWn-IBP in more detail and save discussion at other points of the paper instead.
– Line 202: I need to go back some pages to check how L^\eps is defined again.
– Line 209: in the assumptions, b(x; \theta) is a notation not used before, same for p(x;\theta) to some extent. Also, calling it the probability function is misleading as it may be associated with the data distribution and not the predicted softmax probabilities. The assumptions are also not put “in words” to help understand what they mean. The reader is left alone with thinking about how realistic these assumptions are.
– Line 215: again a different notation for the bias b and the probability p.

- Unstable ReLUs should be explained.
- Equation (5) is also difficult to follow. It would be easier to have a sentence or an additional step to show that the proportionality holds.
- No proof intuition is provided for both theorems. So as reader I only have the choice to spend hours to go through both proofs in detail or not understand where these come from.
- The method is described very briefly and I would not be able to implement it. I think more space should be used to make an accurate description that allows to implement it easily when being familiar with CROWN-IBP.
- In the experiments, the improvement seems small in some cases (e.g., in comparison with IBP for appropriate \epsilon). This is not really discussed. Instead the authors discuss the differences of existing methods depending on \epsilon and \epsilon-scheduling. So from the text it is largely unclear whether the proposed method improves significantly or not.

Conclusion:
Regarding writing, I think that the paper is not ready for NeurIPS. While I think that some insights and the approach can be valuable, the paper is too difficult to follow such that the benefit for the reader is very limited in its current form.

**Time Spent Reviewing:**

2

---

> ### Author Response · Authors · 2021-08-10
> **Authors' response**
>
> We thank the reviewers for their valuable comments (C). Please find our responses (R) for some selected comments below:
>
> - C1: The authors start by referring to “more favorable loss landscape”. However, from the paper it mostly refers to being more smooth. This happens a couple of times throughout the paper and can be confusing – I would prefer being precise and using smoothness throughout the paper.
>     - R1: We say the loss landscape is "favorable", meaning that it is favorable for the optimization
>     - Theorem 1 implies that "the more smooth loss landscape is more favorable for the optimization".
>
> - C2: Regarding the empirical smoothness measures: the authors refer to [29] for computing loss variation. This should be explained in the paper as it seems to be an important point of the paper.
>     - R2: The smoothness measures are detailed in Appendix A.1 as indicated in line 155.
>     - Note that there is a typo in Grad Difference: not "," but "-".
>
> - C3: Line 100: e^{(y)} is not clearly introduced, meaning the reader has to stop and try to understand how exactly C(y) is defined.
>     - R3: $e^{(y)}$ is the standard basis vector with its $y$-th element as 1 and the others as 0.
>
> - C4: Line 120: g^T \cdot is slightly unclear what the \cdot refers to. Generally, in the discussion around Definition 1 it is unclear how exactly g is based on the function bounds. This becomes slightly clearer when discussing IBP, but not to an extent that the paper can be read on its own (without checking related papers).
>     - R4: We write "... $h_i^{[k]}(\cdot)$ can be uppper bounded by a linear function $g^T\cdot+b$ over $\mathbb{B}(x,\epsilon)$".
>     - This is because the variable $x$ is used with different meaning in the same sentence.
>     - We will revise it as "...$h_i^{[k]}(\cdot)$ can be uppper bounded by a linear function $h_i^{[k]}(\xi)\leq g^T\xi+b$ over $\mathbb{B}(x,\epsilon)$".
>
> - C5: Unstable ReLUs should be explained.
>     - R5: It is a relu neuron before which the activation can be both positive and negative.
>
> - C6: In the experiments, the improvement seems small in some cases (e.g., in comparison with IBP for appropriate \epsilon). This is not really discussed. Instead the authors discuss the differences of existing methods depending on \epsilon and \epsilon-scheduling. So from the text it is largely unclear whether the proposed method improves significantly or not.
>     - R6: To summarize, IBP and CBP$\_{1\rightarrow0}$ are much worse ($\Delta$=-7.59,-4.37) than the proposed method in a small epsilon regime ($\epsilon=2/255$) because they lack the tightness, and CBP$\_{1\rightarrow1}$ is much worse ($\Delta$=-4.90) than the proposed method in a large epsilon regime ($\epsilon=8/255$) because CBP$\_{1\rightarrow1}$ lacks the smoothness. The proposed method has the two desired properties: (1) the smoothness of IBP and CBP$_{1\rightarrow0}$ and (2) the tightness of CROWN (CBP$\_{1\rightarrow1}$), without introducing the mixture parameter $\beta$ and it shows a decent performance in a wide range of adversarial perturbations.
>
> - C7: About readability
>     - R7: Thank you for pointing out the readability was insufficient. We tried to understand the phenomenon analytically as well as empirically, which may lack intuitive explanations. We will improve the structure of the paper and provide intuitive explanations with simple notations as you suggest.

---

> > ### Comment · Reviewer_fogS · 2021-08-16
> > **Thanks for the clarifications**
> >
> > Thanks for clarifying some of the raised points. I do not have any follow-up questions/comments at this point.

---

### Decision · Program_Chairs · 2021-09-28

**Decision:**

Accept (Poster)

**Comment:**

The paper is tackling an important question in certified robust training: why tighter bounds may not lead to better models. The authors point out that linear relaxation-based methods may lead to less smooth loss landscapes when upper and lower bounds are not properly chosen, and propose a method to improve linear relaxation-based methods by improving smoothness.

Although the paper is interesting and could be valuable to the certified defense community, there are many presentation problems. Even for expert reviewers who have been publishing in the same area, the paper is very hard to follow. Furthermore, the proposed method only achieves marginal or even no improvement over state-of-the-arts, so whether smoothness is a critical factor in certified training remains shady. We thereby decide the reject the paper this time. We do think the paper contains some interesting ideas and would like to encourage the authors to improve the paper and resubmit to another top conference.

**Consistency Experiment:**

NeurIPS has a long history of experimentation. In 2014, NeurIPS ran an experiment in which 10% of submissions were reviewed by two independent committees to quantify the randomness in the review process. This year, we repeated a variant of this experiment to see how the quality of the review process has changed over time.  This paper was part of the experiment and was therefore assigned to two committees (consisting of reviewers, an Area Chair, and a Senior Area Chair) that reached independent decisions.  If both committees made the same recommendation, this recommendation was followed. If a single committee recommended acceptance, the paper was accepted (with the exception of a few cases in which the other committee identified what we considered a fatal flaw, e.g., an error in a key result).

This copy’s committee reached the following decision: **Reject**

The other committee assigned to the paper recommended **Accept (Poster)**.  You can find the other set of reviews, along with any follow up discussion with the authors here:
https://openreview.net/forum?id=b18Az57ioHn